# Urothelial organoids originating from Cd49f<sup>high</sup> mouse stem cells display Notch-dependent differentiation capacity

Catarina P. Santos [1], Eleonora Lapi[1,2,8], Jaime Martínez de Villarreal [1,2,8], Laura Álvaro-Espinosa[1,8], Asunción Fernández-Barral[2,3], Antonio Barbáchano[2,3], Orlando Domínguez[4], Ashley M. Laughney[5], Diego Megías[6], Alberto Muñoz [2,3] & Francisco X. Real [1,2,7]*

Understanding urothelial stem cell biology and differentiation has been limited by the lack of methods for their unlimited propagation. Here, we establish mouse urothelial organoids that can be maintained uninterruptedly for >1 year. Organoid growth is dependent on EGF and Wnt activators. High CD49f/ITGA6 expression features a subpopulation of organoid-forming cells expressing basal markers. Upon differentiation, multilayered organoids undergo reduced proliferation, decreased cell layer number, urothelial program activation, and acquisition of barrier function. Pharmacological modulation of PPARγ and EGFR promotes differentiation. RNA sequencing highlighted genesets enriched in proliferative organoids (i.e. ribosome) and transcriptional networks involved in differentiation, including expression of Wnt ligands and Notch components. Single-cell RNA sequencing (scRNA-Seq) analysis of the organoids revealed five clusters with distinct gene expression profiles. Together with the use of γ-secretase inhibitors, scRNA-Seq confirms that Notch signaling is required for differentiation. Urothelial organoids provide a powerful tool to study cell regeneration and differentiation.

[1] Epithelial Carcinogenesis Group, Spanish National Cancer Research Centre-CNIO, Madrid, Spain. [2] CIBERONC, Madrid, Spain. [3] Instituto de Investigaciones Biomédicas Alberto Sols, CSIC-UAM and IdiPAZ, 28029 Madrid, Spain. [4] Genomics Unit, Spanish National Cancer Research Centre-CNIO, Madrid, Spain. [5] Weil Cornell Medicine, New York, NY, USA. [6] Confocal Microscopy Unit, Spanish National Cancer Research Centre-CNIO, Madrid, Spain. [7] Departament de Ciències Experimentals i de la Salut, Universitat Pompeu Fabra, Barcelona, Spain. [8] These authors contributed equally: Eleonora Lapi, Jaime Martínez de Villarreal, Laura Álvaro-Espinosa. *email: preal@cnio.es

U rinary bladder diseases, most notably cystitis and bladder cancer, are important medical problems that generate high costs to the health systems worldwide. The bladder, ureters, renal pelvis, and part of the urethra are lined by a multilayered epithelium with some features reminiscent of the skin epidermis. The urothelium consists of three cell types (basal,

intermediate, and umbrella) organized in 3–7 layers, with considerable species-related variation (Fig. 1a). In the mouse, basal cells are small, cuboidal, and express CD44 and KRT5; a fraction thereof express KRT14 and have stem cell properties[1]. Intermediate cells are larger, express KRT5, KRT8, KRT18, and uroplakins UPK1a, 1b, 2, 3a and 3b. Luminal umbrella cells are

**Fig. 1** CD49f labels an organoid-forming urothelial cell population with stem cell features. **a** Urothelial cyto-organization highlighting the three cell layers: basal (CD49f and KRT5), intermediate (KRT5), and luminal (WGA-binding) markers. U, urothelium; LP, lamina propria (scale bar, 50 μm). Color code for the scheme: brown, basal; pink, basal-proliferative; blue, intermediate; green, umbrella. **b** Representative images of organoids from low-passage urothelial cell suspensions embedded in Matrigel in complete medium (upper left). The remaining panels correspond to the leave-one-out experiments (see panel **c**) (scale bar, 500 μm). **c** Quantification of organoid growth in leave-one-out experiments (WR condition: WNT3A and RSPO1 were omitted). Number of organoids normalized to complete medium; error bars indicate SEM. **d**, **e** FACS plots depicting the analysis and isolation of fresh epithelial cells according to cell surface markers (EpCAM, CD49f, CD44, and WGA) and size (n = 2) (scale bar, 500 μm). **f** Quantification of the organoid-forming capacity of freshly isolated and sorted urothelial cells (100 cells/Matrigel drop) compared to the unsorted population (Urothelium); results from a representative experiment (n = 2); error bars indicate SEM. **g** Clonal growth capacity of freshly isolated urothelial cells FACS-sorted according to CD49f expression (CD49f[high] vs. CD49f[low]) and seeded at 1, 10, and 100 cells/Matrigel drop (n = 3). The proportion of Matrigel drops showing outgrowth (bars, Y-axis) and the number of organoids/drop (1, 2, 3, and 4; according to the color code) are shown. **h** Monoclonal origin of organoids established starting from a 1:1 mixture of EGFP- and mTomato-expressing cells (n = 2) (scale bar, 500 μm). ANOVA and Mann–Whitney tests were applied; *p ≤ 0.05, **p ≤ 0.01; ***p ≤ 0.001. Source data are provided as a Source Data file

largest, multinucleated, highly specialized cells expressing high levels of uroplakins and KRT20[2–4]. Umbrella cells constitute the physiological barrier to the passage of water, electrolytes, and urea through tight junctions, responsible for the high resistance paracellular pathway[5]; a role for other urothelial cell types in barrier function has not been demonstrated[6]. Unlike the skin epidermis, the urothelium has a very slow turnover[7] yet it preserves a robust capacity to restore epithelial integrity upon damage[1,8].

Several key transcription factor networks involved in urothelial proliferation/differentiation have been identified using 2D cultures and genetic mouse models. PPARγ is expressed in the urothelium throughout embryonic development and in the adult[9] and it has been shown to participate in proliferation and differentiation, cooperating with FOXA1, KLF5, and EGFR signaling[10–12]. The Southgate laboratory has shown that EGFR inhibition can potentiate the activity of PPARγ agonists and up-regulate the expression of urothelial differentiation markers[13,14]. Retinoic acid signaling also plays an important role in the differentiation of urothelial cells during development and in tissue regeneration upon damage in the adult bladder[15]. However, the role of these pathways in urothelial differentiation is incompletely understood, in part due to the lack of methods to continually propagate normal cells, and improved cellular models are critically required.

In recent years, three-dimensional (3D) organoids have become a powerful tool to study the molecular and cellular basis of epithelial differentiation, allowing consistent culture and perpetuation[16]. Organoids are derived from cells capable of self-renewal and self-organization through cell sorting and lineage commitment in an in vivo-like manner[17]. The Clevers laboratory has pioneered the establishment of organoids from a wide variety of epithelia, including mouse small intestine[16], liver[18], prostate[19], and pancreas[20]. Organoids facilitate studying tissue biology, modeling disease, drug screening, and establishing a solid basis for regenerative medicine and gene therapy[21]. The majority of published studies have focused on organoids derived from simple epithelia. Recently, Lee et al.[22] have reported the establishment of organoids from human bladder tumors and Mullenders et al.[23] have described the features of normal mouse basal organoids and human bladder organoids. However, these reports have not explored in depth the potential of urothelial organoids to understand urothelial biology.

Here, we establish and characterize healthy tissue-derived mouse urothelial organoids and show that high CD49f (integrin α6, ITGA6) expression (CD49f[high]) characterizes a urothelial cell population containing stem cells able to self-perpetuate as organoids. We define their requirements for growth and differentiation and demonstrate their functional properties including barrier formation. Using bulk transcriptomics we identify a role for the Notch pathway in urothelial differentiation. scRNA-Seq allowed us to unveil gene expression signatures featuring Basal, Basal-Proliferative, Intermediate, and Luminal urothelial cells. This analysis further supports that expression of Notch target genes is transiently activated during urothelial differentiation. The organoids described should facilitate and accelerate the study of the molecular pathophysiology of bladder diseases, including the interaction of epithelial cells with pathogens and the mechanisms involved in malignant transformation of the urothelium.

## Results

**Urothelial organoids can be established and perpetuated**. To establish organoids, we isolated cells from digests of urothelial scrapings. This unselected cell population allowed establishing urothelial organoids that could be consistently passaged. To characterize the cell populations present therein, we used flow cytometry analysis with specific antibodies detecting leukocytes (CD45), fibroblasts (CD140a), endothelial cells (CD31), and erythrocytes (Ter119); approximately 20% of single cells lacked these markers. The majority of the cells lacking these markers were EpCAM+ (>60%), indicating their epithelial nature (Supplementary Fig. 1a).

Unselected cells from urothelial digests, or EpCAM+ cells sorted from them, cultured in Matrigel with complete medium (including EGF, LY2157299, Noggin, WNT3A, RSPO1)[21,24] led to growth of multilayered organoids over 1 week. Under these conditions, lumen-containing organoids were very rare (Fig. 1b, Supplementary Fig. 1b). Organoids could be consistently passaged and maintained in culture uninterruptedly for >1 year with stable morphology but a tendency for organoids to become enriched in cells with basal features over time was noted (Supplementary Fig. 1b–d). Unless otherwise indicated, the experiments reported used unsorted urothelial cells and >2 independent organoid cultures at passage <10.

To identify critical growth factors required for organoid formation, we performed leave-one-out experiments where we removed each complete medium component individually or in combination. At day 7, we observed a statistically significant reduction of organoid number upon omission of EGF, WNT3A, RSPO1, or WNT3A + RSPO1 (Fig. 1b, c). As reported for other tissues, Noggin was not required to establish organoids but it was essential for long-term perpetuation[25]. Despite their high proliferative potential, organoids did not form tumors upon xenotransplantation under the skin (Supplementary Fig. 1e) or in the kidney capsule.

**CD49f[high] defines cells with organoid-forming capacity**. To define the cell type of origin of the organoids, we isolated urothelial cell subpopulations based on marker expression and size.

CD49f expression identifies a cancer cell population with basal features[26], suggesting that it might serve as a stem cell marker. Immunofluorescence analysis of normal mouse bladder showed that CD49f is expressed both in the epithelium and in the lamina propria; in the urothelium, CD49f selectively labels basal cells (Fig. 1a)[27]. Compared to CD49f$^{low}$ cells, FACS-purified CD49f$^{high}$ cells were enriched in basal markers whereas CD49f$^{low}$ cells were enriched in luminal markers (Supplementary Fig. 1f). Lectins have also been used effectively as cell-type specific markers. Therefore, we screened a panel of lectins and found that the cytoplasm and plasma membrane of umbrella cells is strongly labeled by wheat germ agglutinin (WGA), while the remaining urothelial cells are weakly labeled (Fig. 1a). FACS-sorting of freshly isolated urothelial EpCAM$^+$ cells on the basis of Cd49f, WGA-binding, and cell size—which augments towards the lumen —showed that CD49f$^{high}$/WGA$^+$ cells (basal) have the highest organoid-forming capacity. By contrast, CD49f$^{low}$ cells (inter-mediate and luminal) were essentially unable to form organoids, regardless of WGA labeling or size (small vs. large) (Fig. 1d). CD44 has also been proposed as an urothelial stem/basal cell marker[3,28]: most CD44$^+$ cells were CD49f$^+$ and both CD49f$^{high}$/CD44$^{high}$ and CD49f$^{high}$/CD44$^{low}$ cells were able to form organoids displaying a similar phenotype (Fig. 1e, Supplementary Fig. 1g, h). However, the number of organoids was highest in the CD49f$^{high}$/CD44$^{high}$ population ($P = 0.029$) (Fig. 1f). Low-density seeding of freshly isolated urothelial cells (1–100 cells/drop) showed that CD49f$^{high}$ cells have a markedly increased clonal growth capacity and that organoids can be established from single cells only from this cell population (Fig. 1g). GFP- and Tomato-labeled sorted cells derived from organoids were mixed prior to seeding in Matrigel at 1:1 ratios and only single-colored organoids were identified, supporting the notion that organoids are monoclonal in origin (Fig. 1h). Altogether, these results indicate that CD49f$^{high}$ and CD44$^{high}$ label the urothelial cell population with highest organoid-forming potential.

**Organoids recapitulate urothelial differentiation and function.** To induce differentiation, organoids that had been maintained for 7 days in proliferation conditions (P) were cultured in medium without EGF, LY2157299, Noggin, WNT3A, and RSPO1 (dif-ferentiation medium, D) for an additional 7 days: a dramatic morphological change was observed, including an increase in diameter and organoid lumen formation, and reduced cell layer thickness (Fig. 2a–d). In proliferation conditions, organoids expressed high levels of Ccne1 transcripts and Ki67 and resemble basal cells expressing Cd49f, Cd44, Tp63, Krt14, and Krt5 and low levels of uroplakins (Fig. 2e–g). By contrast, upon differentiation, organoids showed marked downregulation of cell cycle mRNAs and proteins, a modestly decreased expression of basal markers, and upregulation of mRNA expression of Foxa1 and Pparγ, intermediate cell markers (Krt8 and Krt18), uroplakins (Upk2 and Upk3a), and Krt20 (Fig. 2e–g). The corresponding proteins dis-played the canonical distribution observed in the urothelium: TP63 and CD49f were found in the outer layer of proliferative organoids while PPARγ and UPK3a displayed heterogenous expression in cells lining the lumen of differentiated organoids (Fig. 2f, g). Expression of KRT14 and KRT5 persisted in differ-entiated organoids, possibly reflecting the half-life of these pro-teins and the slow differentiation dynamics of urothelial cells in tissues. KRT20 was generally undetectable at the protein level, as were multinucleated umbrella cells.

Functional competence of organoids was assessed using urothelial barrier assays based on paracellular diffusion of FITC-labeled low molecular weight dextran (FITC-dextran) and fluorescence recovery after photobleaching (FRAP) (Fig. 3a–d).

Urothelial organoids were cultured with medium containing FITC-dextran during both proliferation and differentiation stages. Prior to photobleaching, the lumen of D organoids showed a significantly higher normalized FITC intensity than the lumen of P organoids, suggesting epithelial layer tightness (Fig. 3b, c). After photobleaching, and during a recovery period of up to 14 h, differentiated organoids proved to be impermeable to FITC-dextran whereas proliferative cultures were heterogeneous and contained a mixture of impermeable and permeable organoids (Fig. 3b, d, Supplementary Movie 1). The differences in barrier function acquisition were statistically significant and increased over time of recovery. These findings confirm the ability of organoids to acquire features of differentiated urothelium.

**PPARγ activation and EGFR inhibition enhance differentia-tion.** We used the PPARγ agonist Roziglitazone (Rz) and the EGFR inhibitor Erlotinib to determine if they could further induce urothelial differentiation. P organoids cultured for an additional 7 days in the presence of Rz + Erlotinib acquired larger lumina and showed significantly lower Ki67 labeling, when compared to untreated samples (Fig. 4a, b). Cd49f and Cd44 mRNAs were down-regulated while uroplakin transcripts and proteins were up-regulated (Fig. 4a–c). In D organoids, Rz or Erlotinib alone caused reduced expression of Cd49f and Cd44 mRNAs (Supplementary Fig. 2a). When combined, they led to highest Foxa1 and uroplakin mRNA expression and to a sig-nificant reduction of lumen formation. UPK expression and lumen formation were often, but not always, correlated. There were no major changes in Ki67 and cleaved-caspase-3 labeling upon culture of differentiated organoids with Rz + Erlotinib (Fig. 4a, b). Treatment of organoids with the PPARγ inverse agonist T0070907 at the initiation of the differentiation protocol had minor effects on lumen formation, Ki67, and UPK3a expression (Fig. 4a–c, Supplementary Fig. 2b, c), suggesting that pathways other than PPARγ activation contribute to differentia-tion. KRT20 was not detected in any of the conditions analyzed. These results indicate that PPARγ activation, together with EGFR inhibition, effectively promote urothelial organoid differentiation.

**Cells in differentiated organoids can re-enter the cell cycle.** To determine whether cells could re-enter the cell cycle after indu-cing differentiation, organoids cultured in differentiation med-ium, or in differentiation medium supplemented with Rz and Erlotinib, were maintained for an additional 7 days in complete medium (Fig. 5a–c, Supplementary Fig. 3). In both cases, differ-entiated organoids (D➔P and D + Rz + Erlo➔P, Fig. 5a) acquired a multilayer organization similar to that of proliferative organoids and basal cells showed uniform expression of KI67 and TP63 (Fig. 5c). However, the differences in marker gene expres-sion at the mRNA level were modest, possibly due to pre-existing transcripts (Fig. 5b). The wide re-expression of KI67 in the basal cell layer strongly suggests that incompletely differentiated uro-thelial cells re-enter the cell cycle rather an overgrowth of a low-frequency stem cell population (Fig. 5c). Accordingly, 88.9% of cells in differentiated organoids were CD49f$^{high}$ (Supplementary Fig. 3c). Organoids maintained in differentiation medium (D➔D) showed highest mRNA expression of Foxa1, Pparγ, and uropla-kins; however, smaller lumina and higher cell death were noted (Supplementary Fig. 3b, c). These results indicate that organoids acquire a cyto-organization and differentiation characteristic of the urothelium and retain proliferative potential.

**Transcriptome analysis unveils urothelial differentiation pathways.** To interrogate the transcriptomic programs governing organoid differentiation we performed RNAseq of three

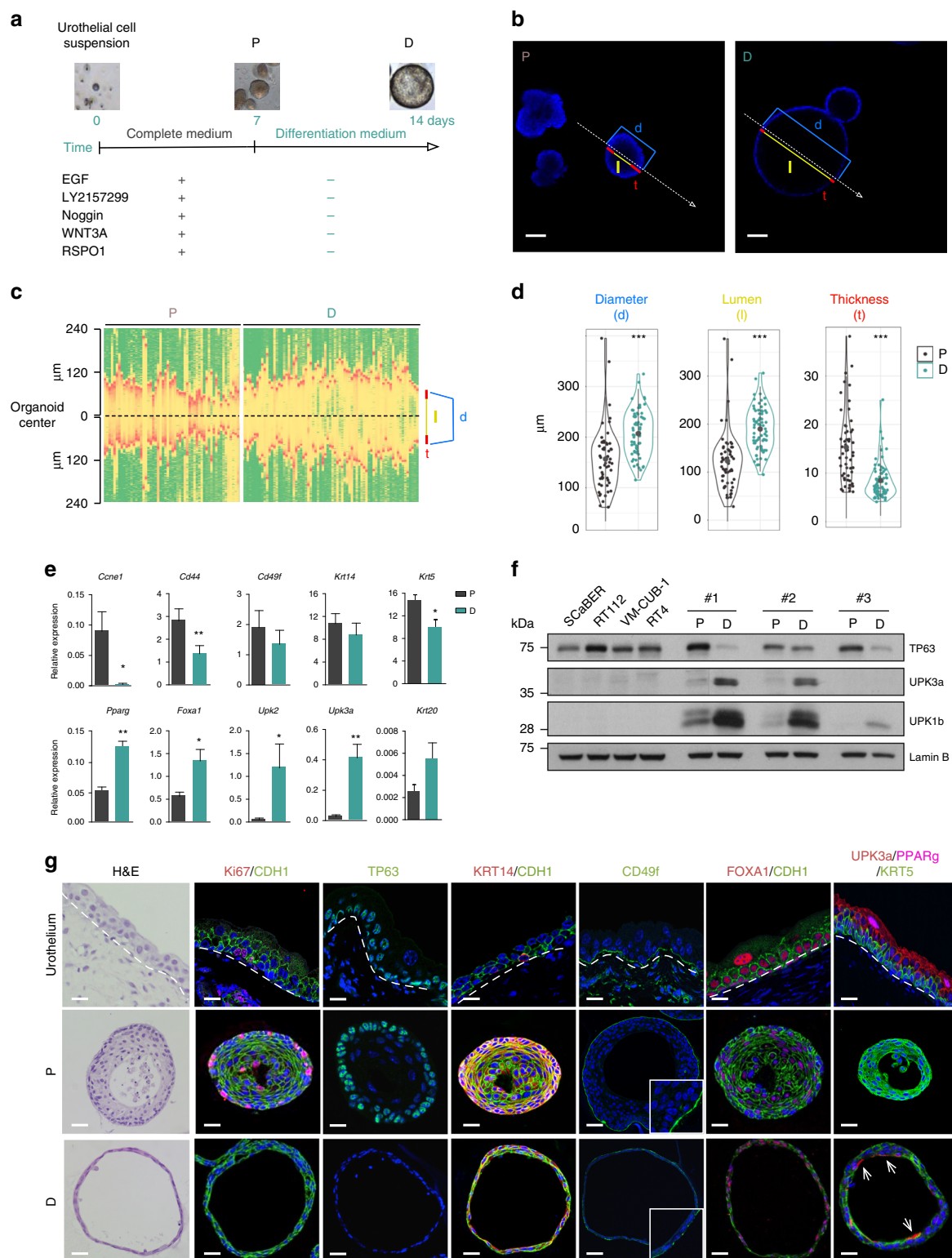

independent paired P and D organoid cultures (Supplementary Fig. 4a). Principal component analysis (PCA) showed that proliferative organoids clustered closely whereas their differentiated counterparts showed greater transcriptome divergence (Supplementary Fig. 4b); 4100 genes were differentially expressed (FDR < 0.05; Supplementary Data 1). Among the top upregulated transcripts in P conditions were those involved in cell cycle (i.e. *Cdk1*, *Aurkb*, *Ccnb2*, and *Ki67*), inhibition of apoptosis (*Birc5*), epidermal differentiation (i.e. *Sprr2f*, *Crnn*, *Stfa3*), cytoskeletal

regulation (*Kif15*, *Kif4*), and stemness (*Cd34*). By contrast, transcripts significantly up-regulated upon differentiation included those involved in urothelial cell functions (i.e. *Upk1a*, *Upk2*, *Upk3a*), glycosylation (i.e. *Wbscr17*, *Ugt2b34*, *Galnt14*), and TGF-β signaling (i.e. *Fstl1*, *Ltbp1*, *Fstl4*) (Supplementary Data 1). Of note, several genes involved in xenobiotic metabolism (i.e. *Cyp2f2*, *Adh7*, *Gstm1*) are among those differentially expressed, underscoring the relevance of this pathway to bladder carcinogenesis[29]. Manual curation revealed the downregulation of canonical basal

**Fig. 2** Growth factor-depleted organoids recapitulate the urothelial differentiation program. **a** Experimental design applied to induce urothelial organoid differentiation: organoids cultured until day 7 in complete medium were maintained for seven additional days in differentiation medium. **b** Image of organoids displaying the features quantified in panel **c**: d, diameter; l, lumen; t, thickness of the epithelial layer. The signal distribution was measured across the organoids as indicated by the arrow in both cases (scale bar, 100 μm). **c** Signal distribution (in microns) acquired by confocal microscopy displaying the quantification of organoid features (diameter) of individual proliferative (P) (n = 57) and differentiated (D) (n = 71) organoids; color code indicates the intensity of the signal: green, low; yellow, intermediate; red, high. **d** Violin plots representing organoid features. **e** RT–qPCR analysis of expression of genes regulated during differentiation. Data are normalized to *Hprt* expression (Mann–Whitney test, error bars indicate SD). **f** Western blot (WB) analysis showing expression of TP63 (basal marker), UPK3a, and UPK1b (luminal markers) in P and D organoids in three independent experiments. Urothelial bladder cancer cell lines (ScaBER, RT112, VMCUB1, and RT4) were used as controls. **g** Immunofluorescence analysis of urothelial markers in P and D organoids. Normal urothelium is shown for comparison. DAPI staining is shown in blue (scale bar, 1000 μm). Source data are provided as a Source Data file

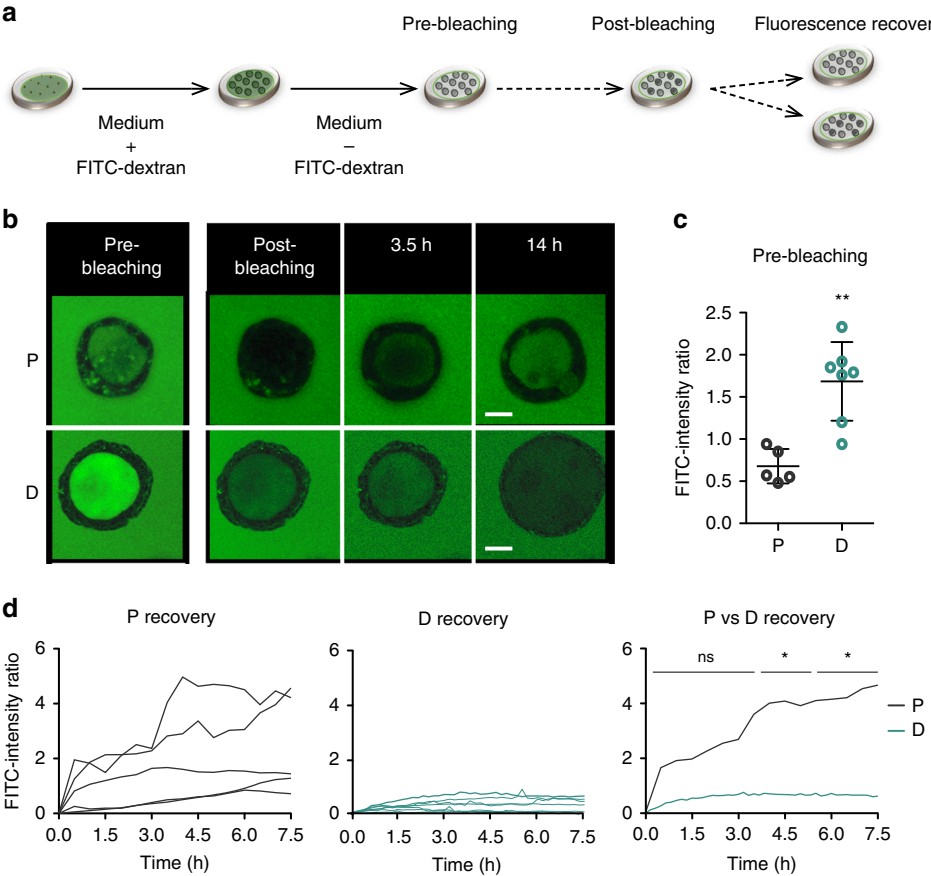

**Fig. 3** Organoids cultured in differentiated conditions are functionally competent and acquire barrier function. **a** Experimental design to assess barrier function in organoid cultures using FITC-dextran and fluorescence recovery after photobleaching (FRAP). **b** Example of P and D organoids during the FRAP assay (pre-bleaching, post-bleaching and recovery—3.5 and 14 h) (scale bar, 1000 μm). **c** Quantification of FITC-dextran intensity of P (n = 5) and D (n = 7) organoids in the pre-bleaching phase showing specific FITC-dextran retention in differentiated organoids (Mann–Whitney test, error bars indicate SD). **d** Fluorescence recovery in P (n = 5) (left) and D (n = 7) organoids (middle); mean fluorescence recovery in P vs D organoids (Mann–Whitney test, error bars indicate SD). *p ≤ 0.05, **p ≤ 0.01. Source data are provided as a Source Data file

urothelial markers (i.e. *Krt14*, *Krt5*, *Cd44*, *Cd49f*, and *Trp63*) and the upregulation of suprabasal markers (i.e. *Krt19*, *Krt8*, *Foxa1*, *Pparγ*, *Upk1a*, *Upk1b*, *Upk2*, and *Upk3a*) in D conditions (Fig. 6a, Supplementary Fig. 4c), as well as the robust regulation of additional keratin species (Supplementary Fig. 4d).

Manual curation also revealed dynamic changes in expression of cell–cell adhesion genes. Transcripts coding for tight junction components showed two distinct expression patterns: *Cldn1*, *Cldn8*, *Cldn12*, and *Cldn25* were down-regulated upon differentiation whereas *Cldn3*, *Cldn4*, *Cldn7*, *Cldn23*, *Ocln*, *Zo-1*, *Zo-2*, and *Zo-3* were up-regulated (Fig. 6b), suggesting distinct functions and cellular distribution for the corresponding proteins. Selected mRNA expression changes were confirmed in independent organoid samples and in normal urothelium

(Supplementary Fig. 4e, f). Apical expression of ZO-1 and CLDN4 was confirmed in differentiated organoids (Fig. 6c). The predicted expression pattern of CLDN1, CLDN3, and CLDN4 was validated using information from the Human Protein Atlas (https://www.proteinatlas.org/). Epithelial impermeability and endocytic traffic were significantly up-regulated in D organoids, in agreement with results of the FITC-dextran assays described above.

Gene set enrichment analysis (GSEA) revealed a highly significant downregulation of the activity of pathways involved in cell cycle, DNA repair, RNA biology, protein synthesis, and cytokines in D organoids; this was accompanied by increased activity of pathways involved in epithelial differentiation/cell–cell adhesion, intracellular traffic (endocytosis, lysosome), and

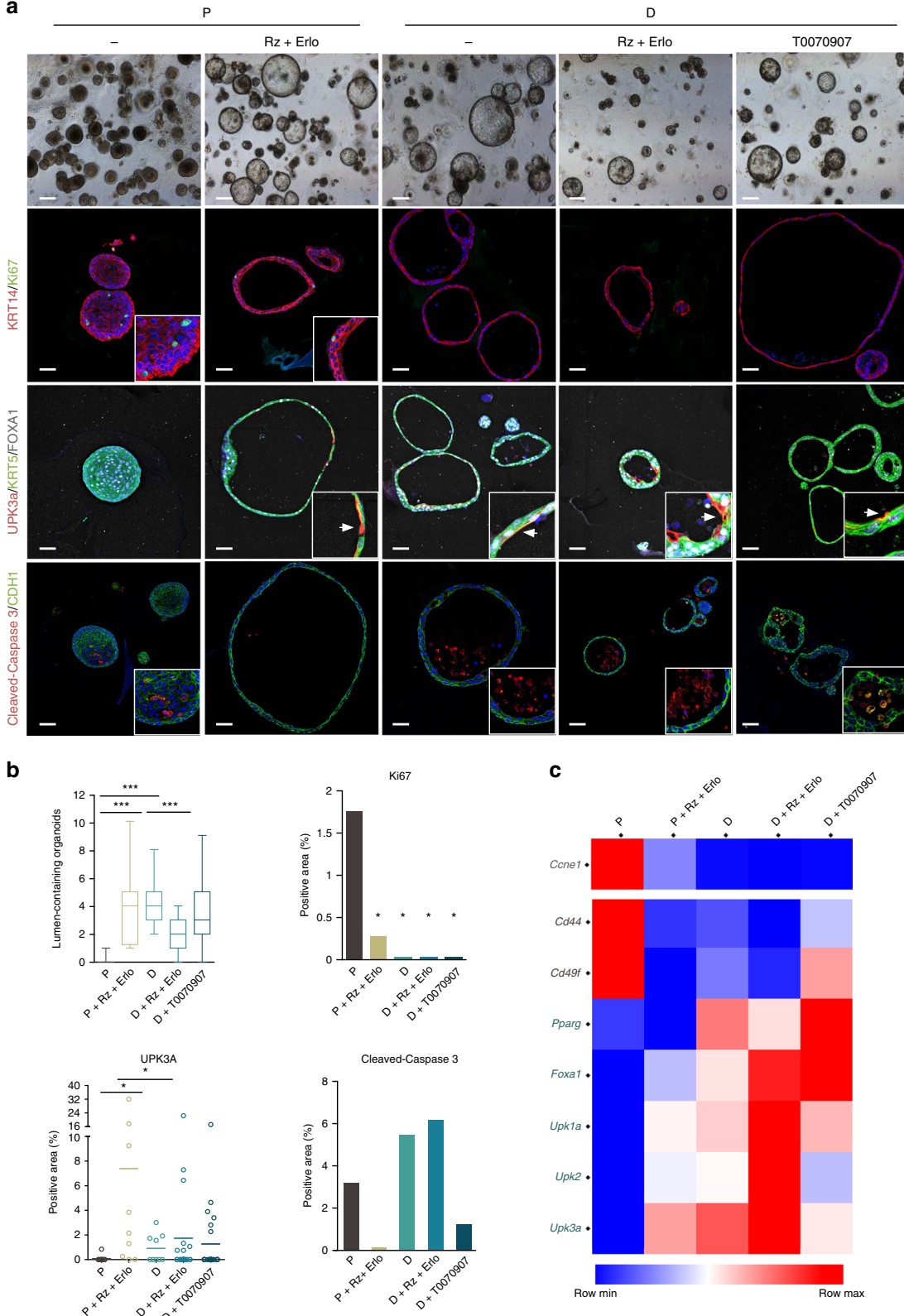

**Fig. 4** Pharmacological modulation of EGFR and PPARγ activity potentiates organoid differentiation. **a** Representative phase contrast and immunofluorescence images of proliferative (P) and differentiated (D) organoids cultured in the presence of drugs modulating PPARγ (Roziglitazone, Rz) or EGFR activity (Erlotinib, Erlo) (scale bars: brightfield, 500 μm; immunofluorescence, 250 μm). **b** Quantification of lumen formation (Mann–Whitney test), Ki67, UPK3a, and cleaved-caspase-3 expression (Bonferroni test) in organoids cultured as described in panel **a**. **c** Heatmap representing RT–qPCR expression analysis of cell cycle and canonical urothelial differentiation markers in P or D organoids treated with Rz + Erlotinib, and with the PPARγ inverse agonist T0070907 ($n = 2$). *$p \leq 0.05$, **$p \leq 0.01$; ***$p \leq 0.001$. Source data are provided as a Source Data file

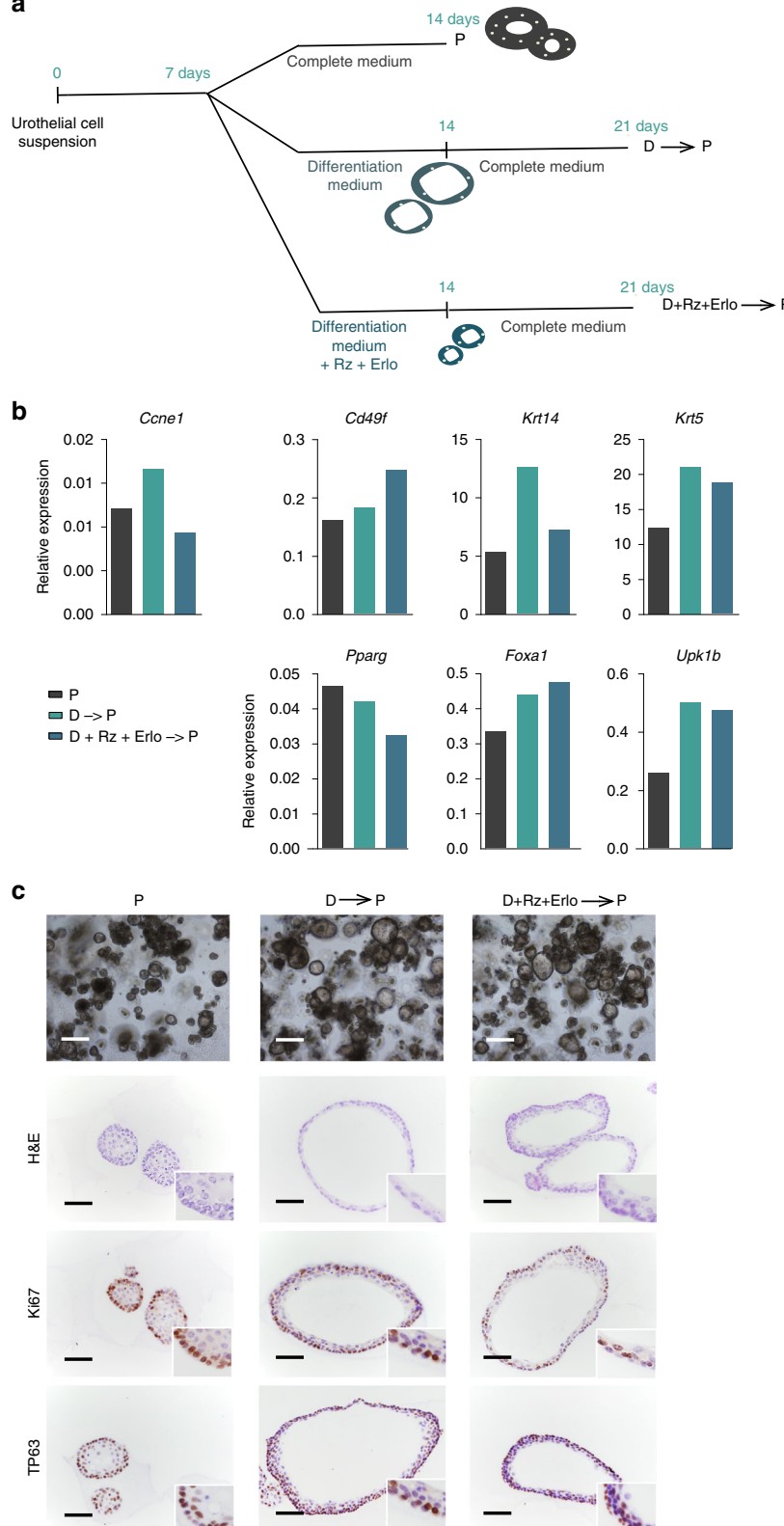

**Fig. 5** Differentiated organoids are able to re-enter the cell cycle upon exposure to complete medium. **a** Experimental strategy: day 7 P organoids were maintained for 7 additional days either in complete medium (P), differentiation medium (D), or differentiation medium supplemented with Roziglitazone and Erlotinib (D + Rz + Erlo). D organoids were then switched to complete medium (D−P; D + Rz + Erlo−P) for 7 additional days. **b** RT–qPCR analysis of expression of genes regulated during differentiation. Data are normalized to *Hprt* expression (*n* = 1 biological replicate). **c** Representative phase contrast images of organoid cultures (scale bar, 500 μm), H&E staining, and immunohistochemistry for Ki67, TP63, and UPK3a (*n* = 1 biological replicate) (scale bar, 250 μm). Source data are provided as a Source Data file

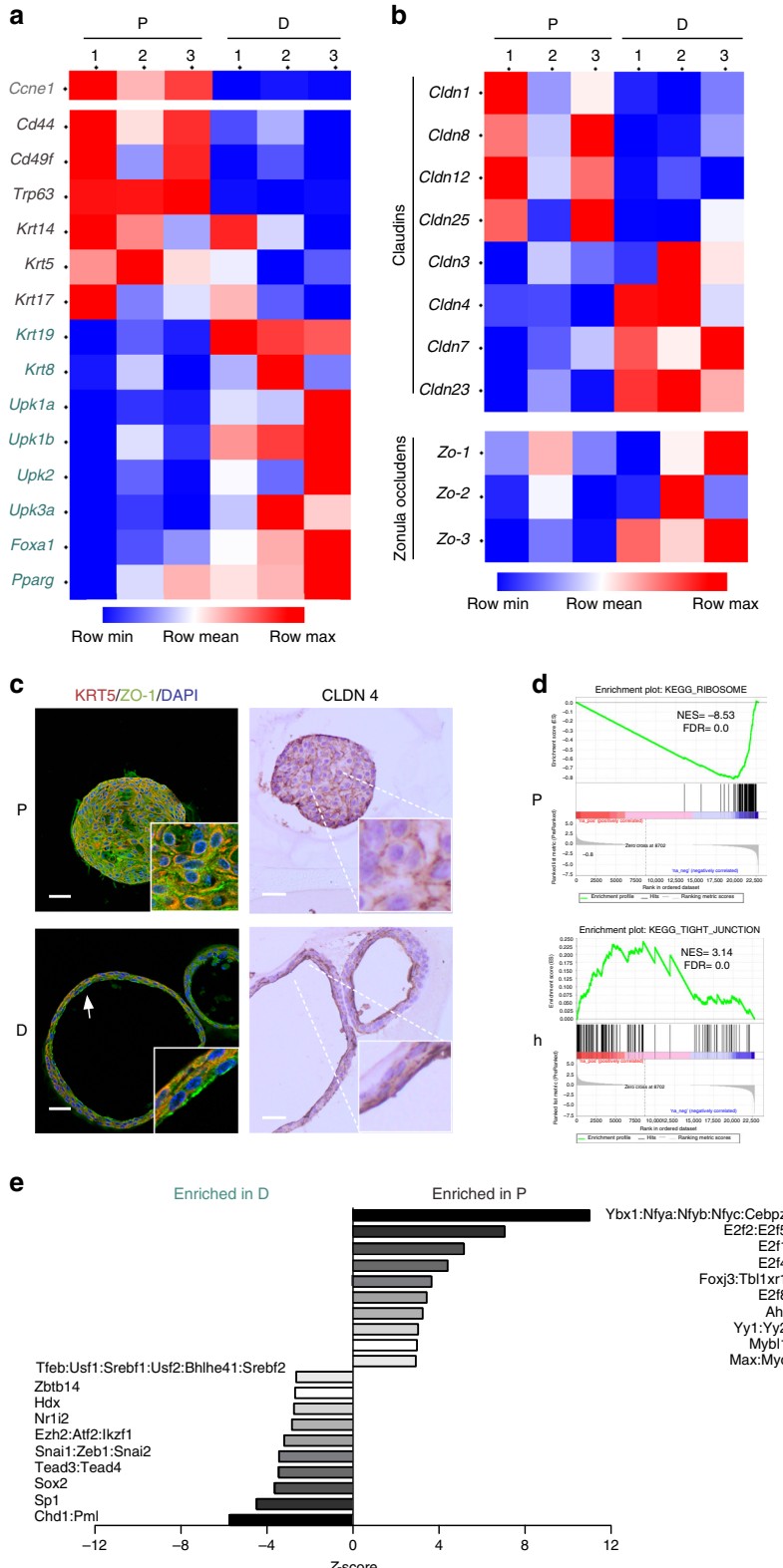

**Fig. 6** Transcriptome analysis reveals organoid differentiation and identifies pathways involved therein. **a** Heatmap showing the expression (FPKM, RNAseq) of key urothelial differentiation genes in P and D organoids (n = 3/group; paired samples). **b** Heatmap showing the expression of genes related to tight junctions (claudins and tight junction/Zo proteins) (FPKM) in P and D organoids (n = 3/group). **c** Immunofluorescence analysis of the expression of ZO-1 and KRT5; immunohistochemical analysis of CLDN4 in the same samples; L (Lumen) (scale bar, 250 μm). **d** Enrichment plots showing the upregulation of ribosome pathway genes and the downregulation of tight junction component genes in P organoids. **e** ISMARA analysis of top transcription factor motifs (ranked by z-scores) significantly enriched in the promoters of genes differentially expressed in P vs. D organoids; for the z-score of motifs enriched in D organoids. Source data are provided as a Source Data file

**Table 1 GSEA showing the top 25 pathways (Kyoto Encyclopedia of Genes) significantly enriched in proliferative and differentiated organoids\***

| Proliferative organoids | Differentiated organoids |
|---|---|
| Ribosome | Lysosome |
| DNA replication | Endocytosis |
| Cell cycle | Focal adhesion |
| Parkinsons disease | Neuroactive ligand receptor interaction |
| Huntingtons disease | Regulation of actin cytoskeleton |
| Oxidative phosphorylation | Phosphatidylinositol signaling system |
| Spliceosome | Melanogenesis |
| Pyrimidine metabolism | Pathways in cancer |
| Alzheimers disease | Tight Junction |
| Purine metabolim | B cell receptor signaling |
| Base excision repair | Axon guidance |
| Mismatch repair | Inositol phosphate metabolism |
| Oocyte meiosis | Other glycan degradation |
| Nucleotide excision repair | Leukocyte transendothelial migration |
| Homologous recombination | Notch signaling pathway |
| RNA polymerase | Fcγ R-mediated phagocytosis |
| RNA degradation | Endometrial cancer |
| Systemic lupos erythematosus | Neurotrophin signaling pathway |
| Proteasome | Wnt signaling pathway |
| N-glycan biosynthesis | Mapk signaling pathway |
| Protein export | Apoptosis |
| Amino sugar and nucleotide sugar metabolism | Adherens Junction |
| Steroid biosynthesis | Acute myeloid leukemia |
| Progesterone-mediated oocyte maturation | Insuline signaling pathway |
| Cardiac muscle contraction | mTOR signaling pathway |

ªAll pathways significant with FDR *q*-value <0.05

signaling (phosphatidylinositol, Notch, Wnt, MAPK, mTOR) (Table 1).

Promoter motif analysis of differentially expressed genes using ISMARA revealed significant enrichment in motifs of TF involved in proliferation (E2F, MYC, MYB) and hypoxia regulation (HIF1A) in proliferative conditions. By contrast, the genes differentially expressed upon differentiation showed enrichment of motifs corresponding to chromatin regulators (CHD1, EZH2, IKZF1), Hippo pathway (TEAD3, TEAD4), and epithelial–mesenchymal transition regulators (SNAI1, ZEB1, ZEB2) (Fig. 6d), pointing to transcriptional networks regulating cell differentiation in the urothelium.

**Wnt and Notch signaling in organoid differentiation**. The pathway analyses suggested roles of the Wnt and Notch pathways in urothelial differentiation. Therefore, we analyzed the effects of drugs targeting both pathways.

The GSK3β inhibitor CHIR99021—acting as a Wnt agonist—had no major effect on organoid morphology when added to proliferative organoids for 7 days. CHIR99021 induced an increase in mRNA expression of the Wnt target genes *Mmp7* and *Axin2*, as well as proliferation markers (*Ccne1* and *Pcna*), consistent with the growth promoting effect of Wnt activation (Supplementary Fig. 5a–c). There were no significant changes in the expression of basal or urothelial markers. Differentiating organoids treated with CHIR99021 displayed a more solid aspect and a significant reduction of lumen formation. There was a similar up-regulated expression of *Mmp7* and *Axin2*, associated

with reduced expression of urothelial markers (Supplementary Fig. 5a–c). To assess the role of endogenous Wnt ligands, we analyzed the effect of the porcupine inhibitor IWP-2 on differentiating organoids: a significant reduction of lumen formation, expression of Wnt target genes, and *Ccne1* mRNA was observed. A modest upregulation of *CD49f* and *CD44* transcripts—without major effects on the expression of urothelial differentiation markers—was detected (Supplementary Fig. 5d–f). Altogether, these data support the notion that the Wnt pathway participates in the maintenance of a progenitor, less differentiated, phenotype.

Regarding Notch, the γ-secretase inhibitor DBZ significantly reduced the expression of the Notch target HES1 at the mRNA and protein levels and completely abrogated lumen formation in differentiating organoids (Fig. 7a–c, e). In these conditions, Notch inhibition resulted in a modest increase in the mRNA expression of basal markers, an upregulation of TP63, and a significantly reduced expression of luminal markers (*Upk1b*, *Upk2*, *Upk3a*, and *Krt20* mRNAs, and UPK1B) (Fig. 7c–f). Similar results were obtained with DAPT, a different γ-secretase inhibitor (Supplementary Fig. 6). Cd49f[high] cells from DAPT-treated differentiated organoids showed increased organoid formation capacity. Altogether, these data support the notion that Notch signaling participates in urothelial differentiation.

**Single-cell organoid RNA-Seq unveils distinct cell populations**. To analyze differentiation programs in the organoids with higher resolution, we performed scRNA-Seq (P organoids, $n = 6826$ cells; D organoids, $n = 4896$ cells). Dimensionality reduction and unsupervised clustering revealed four and three communities in P and D organoids, respectively (Fig. 8a). Detailed results of the analyses are provided in Supplementary Data 2. In proliferative conditions, the clusters displayed features characteristic of Basal-Proliferative, Basal, Intermediate-Low, and Intermediate-High cells. In differentiated conditions, Basal, Intermediate, and Luminal clusters were identified. The Basal-Proliferative cluster was only found in organoids maintained in complete medium whereas the Luminal cluster was exclusive to the differentiated organoids (Fig. 8a, b). The two Basal clusters identified in proliferation conditions shared expression of *Krt14*, *Trp63*, *Itga6/Cd49f*, and *Itgb4* whereas the Basal-Proliferative was selectively enriched in cell cycle genes (Fig. 8c, Supplementary Data 2). The Intermediate-Low cluster was characterized by expression of basal and intermediate (i.e. *Krt8*) markers, transcripts of tight junction components (*Tjp1*, *Cldn4*, *Cldn7*), and known (*Klf5*) as well as candidate (*Foxq1*) urothelial transcription factors. The Intermediate-High cluster showed increased expression of genes up-regulated in the Intermediate-Low cluster and of *Psca*, identified in this study as a marker of urothelial differentiation (Fig. 8c). The genesets enriched in the Basal and Intermediate clusters were similar in proliferative and differentiated organoids. The Luminal cluster showed higher levels of expression of Intermediate cluster markers as well as of uroplakins (Fig. 8c).

Among the differentially expressed genes (DEGs) in the Basal clusters were those coding for several Wnt ligands (*Wls*, *Wnt10a*, *Wnt4*, and *Wnt5a*) (Fig. 8d). Importantly, components of the Notch pathway were also differentially expressed: *Notch1* was enriched in Basal clusters whereas *Jag1* and *Hes1* were enriched in the Intermediate-Low and Intermediate clusters (P and D organoids, respectively) (Fig. 8d). These results suggest that the Notch pathway is transiently activated during urothelial differentiation.

An integrated analysis of both organoid scRNA-Seq datasets yielded five clusters with differentially expressed genes featuring Basal, Basal-Proliferative, Intermediate, Intermediate-High, and

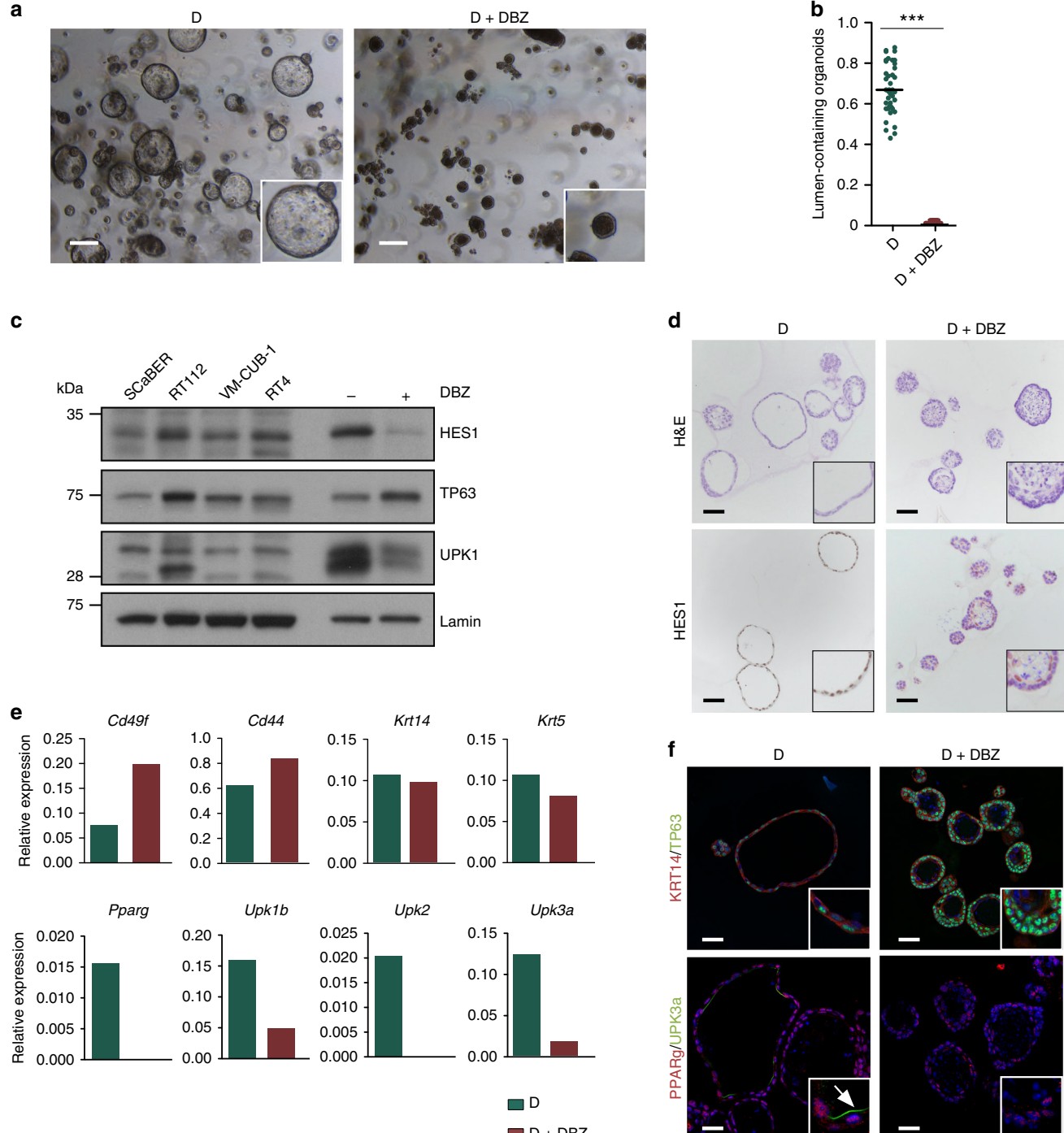

**Fig. 7** Notch pathway inhibition prevents the differentiation of urothelial organoids. **a** Phase contrast images of D organoids cultured in the presence of the γ-secretase inhibition ($n = 4$) (scale bar, 500 μm). **b** Quantification of the effects of DBZ on lumen formation (Mann–Whitney test). **c** Western blot analysis showing the expression of TP63 (basal marker), UPK3a, and UPK1b (luminal markers). **d** H&E and HES1 expression in organoids treated with DBZ (scale bar, 250 μm). **e** RT–qPCR analysis of expression of Notch target genes, proliferation, and urothelial differentiation markers in D organoids treated with DBZ; results were normalized to *Hprt* expression. **f** Immunofluorescence highlighting TP63 upregulation and UPK3a donwregulation upon treatment with DBZ (scale bar, 250 μm). ***$p \leq 0.001$. Source data are provided as a Source Data file

Luminal cells (Fig. 8e, Supplementary Fig. 7c). The DEGs characteristic of each cluster highly overlap with the ones of the individual analysis (Supplementary Fig. 8) suggesting a robust clusterization. Indeed, the Basal and a single Intermediate cluster are largely shared by both organoid populations. The Basal-Proliferative and Luminal clusters were consistently preserved in the integrated analysis (Fig. 8e). Supplementary Fig. 9 shows the expression of selected cell layer markers. Altogether, these findings indicate that the transcriptomes of organoids in proliferation and differentiation conditions share similarities (Basal, Intermediate clusters) but also display unique features (Basal-Proliferative or Luminal clusters).

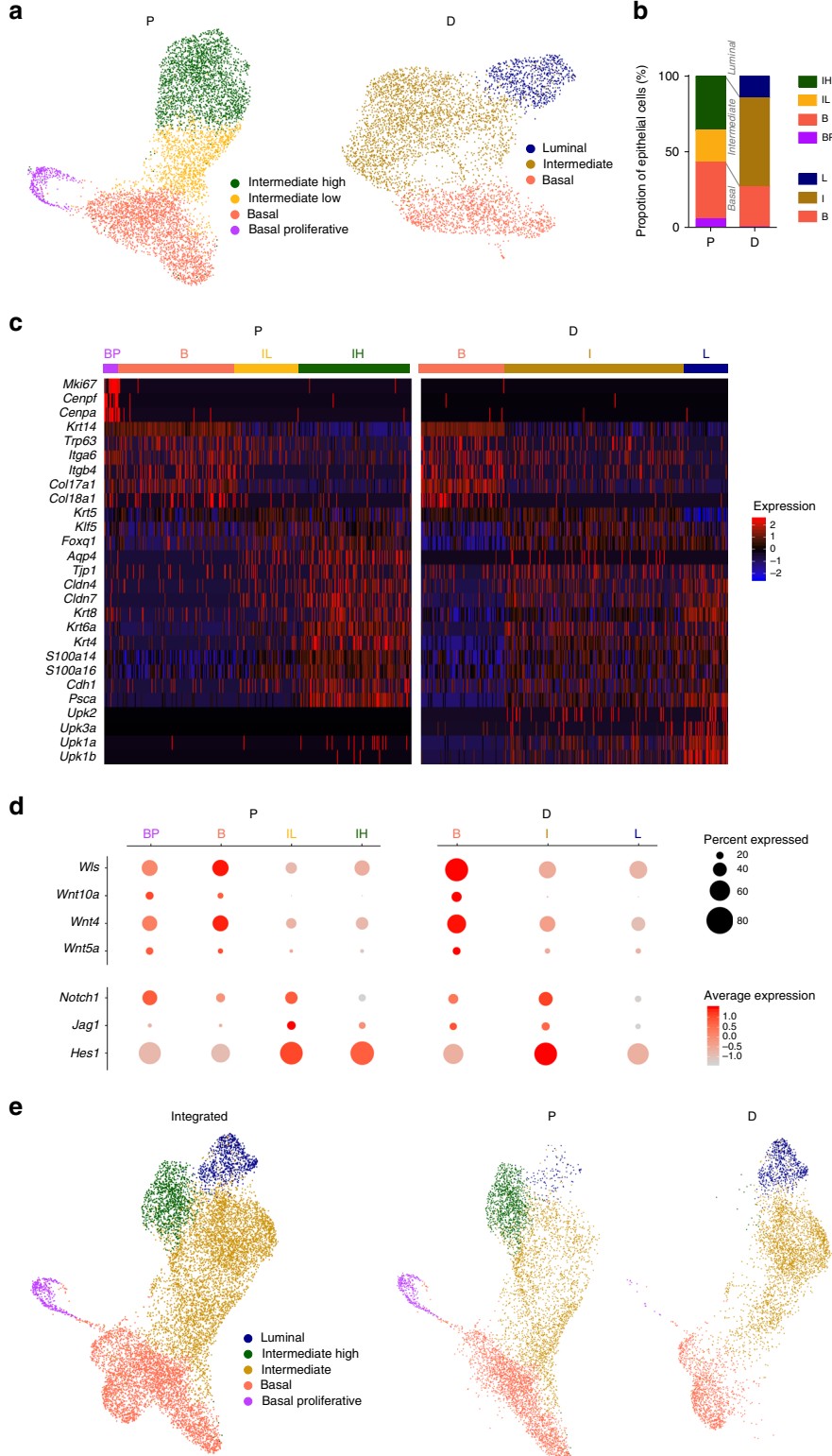

**Fig. 8** Single-cell RNAseq reveals distinct cell populations within P and D organoids. **a** Transcriptomic clusters in P ($n = 6826$ cells) and D ($n = 4896$ cells) organoids visualized using uniform manifold approximation and projection (uMAP) plots, colored according to cluster. **b** Proportion of cells from P and D organoids contributing to each of the clusters shown in panel **a**. **c** Heatmap depicting expression of selected cluster markers identified by differential expression analysis (Wilcoxon rank-sum test). **d** Dot plot depicting the expression of significantly differentially expressed genes from the Wnt and Notch signaling pathways in P and D organoids, according to cell clusters. **e** UMAP plots visualizing integrated analysis of cells from P and D organoids as a joint plot (left, $n = 11722$) or separate plots (P, center; D, right). For P organoids: B (Basal), BP (Basal-Proliferative), IL (Intermediate-Low) and IH (Intermediate-High); for D organoids: B (Basal), I (Intermediate), and L (Luminal). Source data are provided as a Source Data file

## Discussion

The establishment of conditions for the self-renewal and differentiation of urothelial stem cells and the identification of markers thereof are crucial for an improved understanding of homeostasis and dysregulation in disease. In this work, we show that the Cd49f[high] basal urothelial population contains a subset of cells with stem cell properties and long-term growth potential. Upon depletion of growth factors, these cells largely recapitulate urothelial differentiation, allowing to uncover pathways involved therein and providing an important resource for cell biology studies and approaches to regenerative and cancer medicine.

Cd49f/α6 integrin labels a population of urothelial basal cells that is reminiscent of the α2β1 and α3β1 integrin-expressing basal skin epidermal stem cells[30], underscoring that similar hierarchies exist in stratified epithelia despite fundamental differences in proliferation dynamics. The integrated scRNA-Seq analysis revealed that β4 integrin is also significantly up-regulated in the Basal cell cluster (Supplementary Fig. 9). The population of Cd49f[high] cells is heterogeneous and it will be important to identify subsets with highest self-renewal potential.

The growth factor dependencies of Cd49f[high] cells and the growth pathways involved in bladder cancer reveals striking parallelisms. Cd49f and Cd44, highly expressed in proliferative cells with highest organoid-forming capacity, have been proposed as markers of the aggressive basal/squamous-like (BASQ) bladder cancer subtype[27,31,32]. The urothelial organoids reported here display a predominant basal phenotype and are critically dependent on EGF for growth. Similarly, BASQ-type bladder tumors show EGFR pathway deregulation, mainly through EGFR amplification, and are highly sensitive to EGFR inhibitors both in vitro and in xenograft models[33]. Wnt ligands are also required for optimal urothelial organoid growth, consistent with in vivo data indicating that stromal Wnt induced by epithelial Shh is required for urothelial recovery from damage impinged by infection or chemical injury[34]. Addition of the GSK3β inhibitor, acting as a Wnt agonist[35], to organoids prevented the upregulation of urothelial markers in differentiation conditions. scRNA-Seq analysis showed that Wnt ligand mRNAs are selectively up-regulated in cells with basal features and Wnt pathway inhibition reduced the number of lumen-containing organoids, suggesting that epithelial-autonomous Wnt production could contribute to the maintenance of urothelial progenitors. Indeed, inhibition of Wnt secretion led to reduced expression of proliferation markers without major effects on the differentiation signature genes. In bladder cancer, genes involved in Wnt signaling are rarely mutated but epigenetic deregulation and deletion of sFRP1, coding for a Wnt pathway antagonist, has been reported in 25% of tumors[36]. Similarly, a role for Wnt has been proposed on the basis of the cooperation of mutant β-catenin with Pten deletion or HRas activation in mice[37,38]. These findings highlight the similar dependencies of normal and tumor cells across species.

Mullenders et al. have recently reported that EGF was not required for the long-term maintenance of their organoids, indicating that there may exist different urothelial cell populations with specific growth dependencies. In addition, organoid culture in the presence of FGFs resulted in pervasive branching and tube extension, morphological features that are only exceptionally observed in our culture conditions[23].

We show unequivocally that urothelial organoids undergo differentiation as determined by the expression of urothelial signature markers, as well as by global transcriptomic analysis. Growth factor removal led to the downregulation of basal signature genes, including epidermal skin transcripts, and the upregulation of uroplakins and tight junction components. P organoids display a default squamous-like differentiation program while the D counterpart represents a late—but incomplete— differentiation state as indicated by the heterogeneous expression of uroplakins, the lack of morphologically distinguishable luminal umbrella cells, and the paucity/absence of KRT20-expressing cells under the conditions examined. We speculate that the nature of the organoid cell-of-origin, multinucleation, mechanical forces involved in bladder wall distension, and in vitro exposure to urine—among others—may additionally contribute to the formation of umbrella cells and the activation of its characteristic terminal differentiation program[23,39]. Despite the lack of umbrella cells, the transcriptomics analyses provide evidence of the activation of a physiologically relevant gene expression program that results in the formation of organoids displaying impermeability, confirmed by functional assays using FITC-dextran. Interestingly, RNA-Seq unveiled differential expression of transcripts coding for members of the Claudin family in proliferative and differentiation conditions. These findings suggest the contribution of cells other than umbrella cells to the barrier function, although our in vitro studies reported cannot be directly compared with the in vivo situation. In addition, it is conceivable that Claudins—and other tight junction proteins—contribute to other processes beyond cell–cell adhesion[6].

The organoid system described is validated by the response to established regulators of urothelial differentiation such as PPARγ and EGFR inhibitors. Previous studies using 2D cultures had shown that TERT-immortalized urothelial cells lost the ability to respond to PPARγ agonists[40]. By contrast, our organoids undergo differentiation upon culture with PPARγ activators and EGFR inhibitors in complete medium after several in vitro passages. Organoids retained differentiated features in the absence of complete medium components and upon inhibition of PPARγ with an inverse agonist, suggesting either residual PPARγ activity or the participation of additional signaling pathways. In this regard, bulk organoid transcriptomics analysis pointed to a previously unknown role of Notch in normal urothelial differentiation that was confirmed using scRNA-Seq and two different γ-secretase inhibitors. The single-cell analyses revealed highest levels of the Hes1 in cells with intermediate differentiation features, suggesting transient pathway activation during differentiation. These findings are consistent with genetic evidences indicating that somatic mutations in genes coding for Notch pathway components occur in urothelial tumors[41–44] and downregulation of NOTCH1 and DLL1 transcript levels in bladder cancer cells[45]. Notch activation can suppress bladder cancer cell proliferation by direct upregulation of dual-specificity phosphatases; accordingly, ERK1 and ERK2 phosphorylation was associated with NOTCH inactivation and tumor aggressiveness[41]. The findings in human samples are supported by studies using genetic mouse models: ubiquitous or urothelium-specific inactivation of Nicastrin, a γ-secretase complex component, led to BASQ-like tumors and this phenotype was suppressed by inducible overexpression of Notch-IC[41]. In addition, urothelial deletion of Presenilin-1/2 or Rbpj accelerated carcinogen-induced squamous tumors with of epithelial-to-mesenchymal features[44]. These results indicate that Notch is not only involved in tumorigenesis but it also supports differentiation in the urothelium. Inactivation of this pathway is co-opted during carcinogenesis to promote the development of poorly differentiated, aggressive, tumors.

Methods for the expansion of human urothelial tumors as organoids have been reported recently[22,23]. This work shows that the organoids largely reflect the genetic alterations present in the original tumors and they display the phenotype of the major bladder cancer subtypes. They also maintain substantial phenotypic stability while exhibiting plasticity upon in vivo expansion. Furthermore, preliminary evidence indicates that organoids may

predict in vivo drug responses. The most fundamental difference of the culture conditions used in the study of Lee et al.[22] and ours refers to the use of fetal bovine serum. The growth medium used in this study has not allowed the expansion of normal human urothelium or bladder tumors from patients. Further work is required to identify species- and tissue-specific growth requirements, considering the recent reports on human bladder cancer organoids[22,23].

We describe a single-cell transcriptomic analysis of the organoids that not only substantiates the findings at the whole population level but it also provides clues as to the molecular mechanisms involved in urothelial differentiation and homeostasis. For example, cells in the Basal cluster are enriched for expression of several collagen transcripts, suggesting epithelial–mesenchymal features. Intermediate cells are enriched in transcripts for several transcription factors not previously known to be involved in urothelial differentiation, including HOPX and FOXQ1. Interestingly, *FOXQ1* is expressed at highest levels in the bladder (https://gtexportal.org/home/); while rarely altered in tumors, the highest frequency of *FOXQ1* mutations occurs in bladder cancer, with clustering in the vicinity of the forkhead domain (https://www.intogen.org/search). By contrast, *Gata3*, *Foxa1*, and *Ppary* are not differentially expressed among the clusters, pointing to the existence of differentiation regulatory networks with distinct structures. We also identify cell surface markers of these populations that will allow the isolation and characterization of urothelial cell subsets with greater precision. Among them is PSCA, the expression of which is influenced by genetic polymorphisms associated with bladder cancer risk[46,47].

The identification of a mouse urothelial stem cell population able to form organoids with high efficiency, while maintaining differentiation capacity, should facilitate the study of the molecular pathophysiology of bladder diseases, including the interaction of epithelial cells with pathogens. These tools will also allow dissecting the molecular mechanisms through which oncogenes and tumor suppressors contribute to bladder cancer.

## Methods

**Mice**. C57BL/6 (Jackson Laboratories, 000664 RRID:IMSR_JAX:000664), Hsd: Athymic Nude-*Foxn1nu* (Jackson Laboratories, 002019), 129-Gt.*ROSA26 Sor-CAG-EGFPLuo* (Jackson Laboratories, 006053), and *ROSA26 mT/mG* mice (Jackson Laboratories, 007576) were housed in a specific pathogen-free environment according to institutional guidelines. *129-Gt.ROSA26 Sor-CAG-EGFPLuo* mice were kindly provided by Maria A. Blasco (CNIO). Mice were sacrificed by CO$_2$ inhalation. Urothelial bladder suspensions were produced from 7- to 12-week-old mice both for the establishment of organoids and for fluorescence-activated cell sorting (FACS) analyses. For xenograft transplantation, Hsd:Athymic Nude-*Foxn1nu* mice were injected with 10$^6$ cells in 100 μL Matrigel under the flank or 5 × 10$^4$ cells in 50 μL Matrigel under the kidney capsule. All experiments were performed in accordance with the guidelines for Ethical Conduct in the Care and Use of Animals as stated in The International Guiding Principles for Biomedical Research involving Animals, developed by the Council for International Organizations of Medical Sciences (CIOMS), and approved by the institutional Ethics and Animal Welfare Committee (Instituto de Salud Carlos III, Madrid, Spain).

**Cell lines**. ScaBER, RT112, VM-CUB-1, and RT4 urothelial bladder cancer cells were cultured in Dulbecco's modified Eagle's medium (DMEM) (Sigma-Aldrich, D5671), supplemented with 10% FBS (HyClone, Logan, UT, USA), 1% sodium pyruvate (Life Technologies, 11360070), and 1% penicillin/streptomycin (Gibco, 15140122) (basic DMEM). VM-CUB-1 cells were used as a positive control for the xenograft experiments. LWnt3a- producing cells transfected with the Wnt3A cDNA (a kind gift from Dr. E. Battle) were used to produce Wnt3A conditioned medium[48]. RSPO1 was supplemented as the conditioned medium of Expi293 (Life Technologies, A14527) cells transfected with a pOPINE-G-RSPO1-His6 plasmid (a kind gift of Dr. E. Batlle) using the ExpiFectamine™ 293 Transfection Kit (ThermoFisher, A14525). The activity of both conditioned media was confirmed by testing their effect on *Axin2* mRNA expression in HeLa cells using RT–qPCR. Murine macrophage RAW 264.7 cells (a kind gift from Dr. M. Soengas) were maintained in basic DMEM and used to confirm the pharmacological modulation of PPARγ activity. Cells were seeded at 70% confluence in basic DMEM supplemented with oxLDL (15 μg/mL) alone or with either Roziglitazone (10 μM) or

T0070907 (2 μM). After 2 h, cells were cultured with basic DMEM supplemented with oxLDL (15 μg/mL). After 48 h, RNA was extracted. All cells were allowed to settle in a humidified incubator at 37 °C, 5% CO$_2$. All cell lines used were Mycoplasma-free.

**Establishment of mouse organoids**. Mice were sacrificed and the bladder was accessed and turned inside-out leaving the urothelial surface exposed. The urothelium was enzymatically digested with collagenase P (0.5 μg/mL) (Roche, 11215809103) in Hank's Balanced Salt Solution (HBSS) (Life Technologies, 14025050) in a thermoblock with gentle shaking at 37 °C for 20 min. Collagenase P was inactivated with 2 mM EDTA and 50% FBS. The cell suspension was collected and the remaining urothelium was scraped. After filtration through a 70-μm strainer and centrifugation at 1200 r.p.m. for 5 min at 4 °C, the pellet was washed 2× with washing medium [Advanced DMEM F12 medium (Gibco, 12634010) + 1× HEPES (Gibco, 15630080) + 1× Glutamax (Gibco, 35050061)] and cells were embedded in growth factor-reduced and phenol red-free Matrigel (Corning, 356231). Matrigel-cell suspensions (20 μL drops) were plated onto six-well plates, allowed to settle in a humidified incubator at 37 °C/5% CO$_2$, and overlaid with 2 mL complete medium (see below). Medium was replaced every 2–3 days. Cultures were usually expanded at a 1:4–1:6 ratio every 7–10 days. After Matrigel removal with Cell Recovery Solution (Corning, 354253) on ice, the cell suspension was washed with phosphate-buffered saline (PBS), then with washing medium, and centrifuged at 1200 r.p.m. for 5 min at 4 °C. Then, organoids were chemically digested with Dispase II solution (10 mg/mL) (Gibco, 17105041) for 15–20 min in a rotating wheel at room temperature. The reaction was neutralized with 2 mM EDTA and single cells were obtained by mechanical disruption with a syringe with a 21G needle for at least strokes. After two washes with washing medium, the pellet was embedded in fresh ice-cold Matrigel, seeded on six-well plates, and covered with medium as described. Unless otherwise specified, organoids were maintained in culture with complete medium [Advanced DMEM/F12, 1× penicillin/streptomycin, 1× HEPES, 1× GlutaMAX, 50% WNT3A conditioned medium, 5% human RSPO1 conditioned medium, 1× N2 (Gibco, 17502048), 1× B27 (Gibco, 12587010), 50 ng/mL human recombinant EGF (Invitrogen, PHG0311L), 1 mM *N*-acetylcysteine (Sigma-Aldrich, 616-91-1), 50 μg/mL human Noggin (Peprotech, 120-10C) and 1 μM LY2157299 (AxonChem, 700874-72-2)]. The ROCK inhibitor Y-27632 (10 μM) (Sigma-Aldrich, 129830-38-2) was added during the first 3 days of culture. For differentiation experiments, organoids were cultured for the first 7 days in complete medium, reseeded (without disaggregation) in fresh Matrigel, and cultured either with complete or differentiation medium (lacking WNT3A and RSPO1 conditioned medium, EGF, LY2157299, and Noggin) for the following 7 days. PPARγ modulators were added to the medium (complete or differentiated) as indicated in the text, at the following concentrations: Rosiglitazone (1 μM) (Sigma-Aldrich, 122320-73-4), T0070907 (10 μM) (Sigma-Aldrich, 313516-66-4). The EGFR inhibitor Erlotinib was used at 0.5 μM (Sigma-Aldrich, CDS022564). The γ-secretase inhibitors DAPT (5 μM) (Sigma-Aldrich, 208255-80-5) or DBZ (2.5uM) (Tocris, 4489), porcupine inhibitor IWP-2 (5 μM) (Merk, 681671), and Wnt agonist CHIR99021 (2.5 μM) (Tocris, 4423) were added to P organoids on day 7 and organoids were allowed to grow in differentiation medium until day 14. Microphotographs were taken using an inverted microscope (Olympus CK-30). In order to cryopreserve organoids, 7-day cultures were washed once in PBS, Cell Recovery Solution was added, and cells were collected and placed on ice for 30 min. After washing once with PBS, then with washing medium, and centrifugation at 1200 r.p.m. 4 °C for 5 min, organoids were resuspended in freezing medium [10% DMSO (Sigma-Aldrich, D2650) in Advanced DMEM/F12 supplemented with 10 μM Y-27632] at a density equivalent to three confluent drops/500μL. Cryovials were stored at −80 °C. For thawing, vials were placed in a 37 °C water bath and the contents washed twice with Advanced DMEM/F12 before reseeding in Matrigel at the required density.

In experiments analyzing FACS-sorted populations, cell viability of the different populations was assessed and found to be comparable.

All experiments reported were performed without cell sorting based on EpCAM expression, using low-passage cultures (<10), unless stated otherwise.

For all major experiments, at least two—and generally three—independent biological replicates—in addition to technical replicates—were performed.

**Flow cytometry analysis**. Urothelial cell suspensions obtained as previously described were incubated with blocking buffer (1% bovine serum albumin (BSA)/ 3 mM EDTA in PBS) for 15 min at room temperature. After washing twice with PBS, cells were incubated with FITC-labeled anti-mouse EpCAM (BioLegend, 118207), APC-labeled anti-mouse EpCAM (BioLegend, 118213), FITC-labeled WGA (Vector Laboratories, FL-1021), PE-labeled anti-mouse/human CD49f (BD Biosciences, 555736), APC-labeled anti-mouse CD45 (BD Biosciences, 559864, PE-labeled anti-mouse CD31 (BD Biosciences, 555027), PE-labeled anti-mouse CD140a (Labclinics, 16-1401-82), PE-labeled anti-mouse Ter119 (BioLegend, 116208), and/or APC-labeled anti-mouse/human CD44 (BioLegend, 103011) antibodies in FACS buffer (0.1% BSA/3 mM EDTA in PBS) for 30 min at 4 °C. After washing twice with PBS, cells were resuspended in FACS buffer and stained with DAPI (Sigma-Aldrich, D9542). In all the experiments a control sample lacking primary antibody and a Fluorescence Minus One (FMO) control were used. In the experiments using isolated urothelial cells, EpCAM + single cells were sorted by

FACS and dead cells were excluded from subsequent analyses. In the experiments with organoids, samples were disaggregated as previously described and single-cell suspensions were incubated with PE-labeled anti-mouse/human CD49f antibody. In the case of $D < 70\,\mu m$ and $D > 70\,\mu m$, organoids cultured for 7 days in complete medium were filtered using a 70 μm filter and the fall through and the retained organoids were reseeded in Matrigel and cultured in differentiation medium. All samples were analyzed using a FACS Influx or AriaII (BD Biosciences) flow cytometer and at least 10,000 events were acquired. Analyses were performed using FlowJo version 10.2 flow cytometer analysis software.

**Clonality experiments**. FACS-sorted freshly isolated urothelial cells were embedded in 5 μL of Matrigel in a 96-well format at 1, 10, or 100 cells/well for the clonal growth experiments. For the monoclonality experiment, organoids derived from mice of the 129-Gt.ROSA26 Sor-CAG-GFPLuc and ROSA26 mT/mG;Ubc-CreERT2 strains were separately established; after dissociation to single-cell suspensions they were mixed at a 1:1 ratio and allowed to grow as organoids in 20 μL of Matrigel/drop in 48-well plates at 1000 cells/well. Organoids containing cells of the corresponding fluorescent color (EGFP, Tomato, and chimeras) were counted.

**Cell cycle re-entry experiments**. Urothelial cell suspensions were seeded in complete medium for 7 days. Afterwards, organoids were resuspended in Matrigel and cultured for the following 7 days either in complete medium, differentiation medium, or in differentiation medium supplemented with Rosiglitazone (1 μM) and Erlotinib (0.5 μM). On day 14, the medium was replaced with either complete or differentiation medium and cultures were allowed to grow until day 21.

**Organoid quantification**. For the leave-one-out experiments, images were acquired with ×40 resolution with CCD-microscope using a brightfield filter. Three pictures in the Z-axis were taken in order to collect the majority of the organoids. Then, a Z-stack was done using ImageJ software. For immunofluorescence and growth assays, quantification was performed with tailored routines programmed in Definiens XD v2.5 software. Data on organoid features (layer thickness, lumen, and diameter) were derived from the signal distribution upon Hoechst staining. Organoid growth and time-lapse videos were acquired using a DMI6000B brightfield microscope from Leica Microsystems with an HC PL APO ×10 0.4 NA dry objective. Cells were maintained in a temperature-controlled (37 °C), humidified environment in the presence of 5% $CO_2$ during imaging.

**Barrier function assays**. A derivative of FRAP-derived strategy was used whereby the influx of Fluorescein isothiocyanate–dextran (average mol wt 3000–5000) (Sigma-Aldrich, 60842-46-8) into the organoid is measured after bleaching the original fluorescence. Organoids were cultured in μ-slide eight-well Ibidi plates (Ibidi, 80826). In the pre-bleaching step, the original signal was registered. After bleaching by pointing the laser into the lumen for 20 s (488 nm wavelength at maximum power), the signal decayed and a new register was established. In the recovery phase, the entrance of FITC-dextran into the lumen was measured. The recovery period spanned for a total of 14 h, with images acquired every 30 min. The assay was performed using a Leica TCS-SP5 (AOBS) Leica Microsystem laser scanning confocal microscope with a ×40 immersion oil objective (HCX PLAPO 1.2 NA). The normalized intensity of luminal FITC-dextran was calculated by dividing the measurements corresponding to the region of interest in the lumen by the background in the Matrigel.

**Western blotting**. Organoids were pelleted after removing Matrigel with Cell Recovery Solution and whole cell extracts were prepared using RIPA buffer [0.05 M Tris-HCl pH 7.5, 0.1% SDS, 0.15 M NaCl, 1% Triton X-100 (Sigma-Aldrich) and 1% sodium deoxycholate] containing protease and phosphatase inhibitors (leupeptin, 10 μg/mL; aprotinin, 10 μg/mL; PMSF, 1 mM; orthovanadate, 1 mM; NaF, 1 mM; all from Sigma-Aldrich). Cell lysates were fractionated by SDS-PAGE, transferred to PVDF membranes, and incubated with the following primary antibodies: mouse monoclonal anti-UPK3a (Santa Cruz, sc-166808), mouse monoclonal anti-UPK1b (a kind gift from Dr. A. García-España), goat polyclonal anti-Lamin (Santa Cruz, sc-6216), rabbit monoclonal anti-HES1 (a kind gift from Dr T. Sudo), mouse monoclonal anti-TP63 (Abcam, ab735). Enhanced chemiluminescence was used; films were exposed to ensure that bands were not saturated.

**Immunofluorescence and immunohistochemistry**. Matrigel drops containing organoids were collected 7 days after seeding, fixed in 10% formalin, and embedded in paraffin. Mouse bladders (C57BL/6) were embedded in OCT or formalin-fixed and paraffin-embedded. Sections (3 μm) were obtained for immunofluorescence and immunohistochemistry analyses. After deparaffinization and rehydration, antigen retrieval was performed by boiling in citrate buffer pH 6 for 10 min. For IF, sections were incubated with 3% BSA/0.1% Triton in PBS for 45 min at room temperature and incubated with primary antibodies overnight at 4 °C. After washing with 0.1% Triton/PBS, the appropriate fluorochrome-conjugated secondary antibodies (Alexa Fluor 488-labeled goat anti-mouse Ig (A-11001), Alexa Fluor 555-labeled goat anti-rabbit Ig (A-21428), Alexa Fluor 488-labeled goat anti-chicken Ig (A-11039), and Alexa Fluor 680-labeled goat anti-rabbit Ig (A-27042);

Life Technologies) were added for 45 min, sections were washed, and nuclei were counterstained with DAPI. After washing with PBS, sections were mounted with Prolong Gold Antifade Reagent (Life Technologies, P36930). Images were acquired using a confocal microscope (Leica, SP5) with a ×40 immersion oil lens and a zoom of 1–2.5. For immunohistochemistry, endogenous peroxidase was blocked with 3% $H_2O_2$ in PBS for 30 min, sections were incubated with 3% BSA/0.1% Triton in PBS for 45 min at room temperature, and with primary antibody overnight at 4 °C. After washing with 0.1% Triton/PBS, sections were incubated with the appropriate secondary antibodies (Envision kit for mouse K4001 or rabbit K4003 Ig, Dako) for 45 min, washed, and reactions were developed with DAB (Dako, K3468). Sections were lightly counterstained with hematoxylin, dehydrated, and mounted. All shown images are representative of >10 random fields.

The following reagents were used: FITC-labeled WGA (Vectorlabs, FL-1021), PE-labeled anti-mouse/human CD44 (BioLegend, 103023), PE-labeled anti-mouse/human CD49f (BD Biosciences, 555736), mouse monoclonal anti-Claudin4 (Santa Cruz, sc-376643), rat monoclonal anti-Zo-1 (Santa Cruz, sc-33725), rabbit monoclonal anti-PPARγ (Cell Signalling, 2435T), rabbit polyclonal anti-Ki67 (Novocastra, NCL-Ki67p), rabbit monoclonal anti-FOXA1 (Abcam, ab173287), mouse monoclonal anti-E-Cadherin (BD Biosciences, 610181), rabbit monoclonal anti-Cleaved-caspase-3 (Cell Signalling, 9579), rabbit polyclonal anti-KRT5 (BioLegend, 905501), rabbit polyclonal anti-KRT14 (BioLegend, 905301), anti-TP53, mouse monoclonal anti-UPK3a (Santa Cruz, sc-166808), mouse monoclonal anti-UPK1b (kind gift from Dr. A. García-España), rabbit monoclonal anti-ITGA6/CD49f (Abcam, ab181551), and rat monoclonal anti-HES1 (395A/A7, CNIO Monoclonal Antibody Core Unit). Antibody concentrations are provided in the Nature Protocol Exchange article[49] and Reporting summary.

**Real-Time quantitative PCR**. Total RNA was extracted from organoids using the TRI Reagent® (Sigma-Aldrich, T9424) followed by the PureLinkTM RNA Mini Kit (Life Technologies, 12183020), according to the manufacturer's instructions. For cells grown in 2D culture, RNA was extracted using ReliaPrepTM RNA Cell Miniprep System Kit (Promega, TM370). Samples were treated with DNase before reverse transcription (Life Technologies, AM1906). cDNA was generated from 200 to 1000 ng of RNA using random hexamers and reverse transcriptase using the TaqMan® Reverse Transcription Reagents (Life Technologies, N8080234). Reaction mixes lacking RT were used to ensure the absence of genomic DNA contamination. PCR amplification and analyses were conducted using the 7900HT Real-Time PCR System (Applied Biosystems, Life Technologies) using GoTaq® qPCR Master Mix (Promega, TM318). Gene-specific expression was normalized to *Hprt* (organoid cultures), *β-actin* (RAW 264.7 cells), and *GAPDH* (HeLa cells) using the ΔΔCt method. For RT–qPCR, primer pairs were designed to achieve inter-exon products of 200–250 bp. Primer sequences are provided in Supplementary Table 1.

**Bulk RNA-sequencing analysis**. RNA quality was assayed by laboratory chip technology on an Agilent 2100 Bioanalyzer. PolyA + RNA was isolated from total RNA (1 μg, RIN > 9), randomly fragmented, converted to double stranded cDNA, and processed through subsequent enzymatic treatments of end-repair, dA-tailing and ligation to adapters according to Illumina's TruSeq RNA Sample Preparation v.2 Protocol. The adaptor-ligated library was completed by limited-cycle PCR with Illumina PE primers (8 cycles). The resulting purified cDNA library was applied to an Illumina flow cell for cluster generation (TruSeq cluster generation kit v.5) and sequenced on a Genome Analyzer IIx with SBS TruSeq v.5 reagents, according to the manufacturer's protocols. The sample sequencing was paired-end. Three paired samples of organoids cultured in proliferative and differentiation conditions were analyzed. The Nextpresso version 1.9.1 analysis pipeline (Bioinformatics Unit, CNIO, Madrid) was used to process the data with the version MGSCv376/mm8 of the mouse genome[50]. Promoter motif analysis of differentially expressed genes was performed using ISMARA version 1.2.1.

**Principal component analysis**. The Pearson correlation was calculated from the expression values (expressed as fragments per kilobase of transcript per million mapped reads) of each gene for each sample by using the "cor" command in R (https://www.r-project.org/). Principal component analysis was performed using the "prcomp" command in R, from the correlation value of each sample.

**Gene Set Enrichment Analysis**. The list of genes was ranked by the "t-stat" statistical value from the cuffdiff output file. The list of pre-ranked genes was then analyzed with GSEA for Gene Ontology (GO) database. Significantly enriched GO terms were identified using a false discovery rate *q* value of less than 0.25. The analyses were carried out as defined in http://www.broadinstitute.org/gsea/doc/GSEAUserGuideFrame.html?Interpreting_GSEA.

**Droplet based single-cell mRNA sequencing**. Organoids were enzymatically dissociated and cells were suspended in PBS with 0.04% BSA (Ambion AM2616). Cells (10,000 per condition, cell viability >70%) were loaded onto a 10x Chromium Single Cell Controller chip B (10x Genomics) as described in the manufacturer's protocol (Chromium Single Cell 3′ GEM, Library & Gel Bead Kit v3, PN-1000075). Generation of gel beads in emulsion (GEMs), barcoding, GEM-RT clean up, cDNA amplification, and library construction were performed following the

manufacturer's recommendations. Libraries were loaded at a concentration of 1.8 pM and sequenced in an asymmetrical pair-end format, with 28 bases for read 1 and 56 for read 2 in a NextSeq500 instrument (Illumina). Sequencing depth was 58 and 53 million paired reads for proliferative and differentiated organoids, respectively.

**Single-cell RNAseq data computational analysis**. Reads were locally processed with Bcltofastq (bcl2fastq 2.19.0; Illumina). Cell Ranger version 3.0.2 software (10x Genomics) pipeline was then used to demultiplex and align reads to the GRCm38/mm10 transcriptome. FastQC software was used to check sequencing read quality. Cell Ranger count generated the matrices used in the next analysis step.

Cell Ranger matrix data (barcodes, features, and count matrix) were loaded onto the Seurat R package (version 3.0.0.9000). In order to identify and exclude low-quality cells from downstream analyses pre-processing was performed for both datasets. The distribution of UMIs (Unique Molecular Identifiers), genes, and the percentage of mitochondrial-encoded genes across cells were visualized using the ggplot2 R package. Cells with >800 genes detected were kept. Since a high percentage of mitochondrial genes is considered a characteristic of low-quality cells[51], cells expressing >12% of mitochondrial genes were excluded from downstream analysis (Supplementary Fig. 9). After the filtering steps outlined above, 6826 cells with a median gene count of 1151.5 were retained for downstream analysis of the proliferative organoids and 4896 cells with a median gene count of 1346.5 were retained for downstream analysis of the differentiated organoids.

Normalization of filtered cells was performed by dividing gene counts for each cell by the total number of counts for that cell multiplied by a scale factor (10,000-default parameter) and then natural log-transformed.

To identify genes exhibiting high variability across cells in each dataset, and use those heterogeneous features in downstream analysis, we used "vst" method implemented in Seurat v3 FindVariableFeatures function[52]. This method performs feature selection by computing the mean and variance of each gene using unnormalized data and applying log 10 transformation to both and then fitting a line to the relationship of log(variance) and log(mean) using local polynomial regression (loess). Standardized feature values are obtained with the observed mean and the expected variance, given by the fitted line. Feature variance is then calculated on the standardized values after clipping to a maximum value (default parameter: square root of the number of cells). This variance is used to rank the features for each dataset. A subset of 2000 genes (Supplementary Fig. 7b, Supplementary Data 2) exhibiting high cell-to-cell variation was selected for downstream analysis. Lastly, genes were scaled and centered in each dataset by z-normalization to standardize the dynamic range across genes. This was done to ensure subsequent downstream analysis was not biased towards highly expressed genes.

Linear dimensional reduction was performed on the scaled data using principal component analysis (PCA). Elbow plots and JackStraw permutations test were applied to determine the dimensionality of each dataset as significant principal components ($p$ value cut-off of less than 0.05) (25 PCs for proliferative and 21 PCs for differentiated organoids).

The minimal number of clusters was estimated by determining the robustness of the consensus matrix using SC3 R package[53]. The identification of biologically relevant communities was performed by means of a graph-based clustering in Seurat through the construction of a KNN graph and posterior cell clusterization using the Louvain algorithm (resolution 0.2). Cell clusters were visualized using uniform manifold approximation and projection (UMAP)[54] plots with previously selected significant components as an input.

Cluster gene markers were detected with Seurat R package using a Wilcoxon rank sum test between each cluster and the rest of cells in the dataset and $p$ value adjustment was performed using Bonferroni correction based on the total number of genes in the dataset.

Both organoid datasets were integrated into a shared space through the identification of common shared features termed anchors between cells across P and D organoids[52]. Both datasets were pre-processed and normalized separately as described above. Dimensional reduction was performed by means of canonical correlation analysis (CCA)[55]. We defined the dimensionality of the integrated dataset by using 25 canonical vectors CCV (maximum number of significant principal components chosen in individual analysis) that project the two datasets into a correlated low dimensional space. After dimensional reduction, we identified the K nearest neighbors (KNNs) for each cell within both datasets, using the L2 normalized CCV. Lastly, we identified the "anchors" as pairwise correspondences by mutual nearest neighbors (MNN). These anchors were filtered and scored to reduce the effect of incorrect identification and weighted in a matrix that defines the strength of association of each cell and each anchor. Batch correction was then performed by matching MNN as described by Haghverdi et al.[56]. These steps were implemented in the FindIntegrationAnchors and IntegrateData Seurat functions. Downstream analysis on the integrated dataset was performed as it was done for individual analysis (Clustering (resolution 0.17), Cluster markers, and differential expression analysis).

**Additional resources**. Claudin expression data were extracted from the Human Protein Atlas (www.proteinatlas.org). GenePattern was used to compute all heat-maps here presented. IDraw was used to generate the panels and figures.

**Quantification and statistical analyses**. All quantitative data are presented as mean ± s.e.m. from ≥2 experiments or samples per data point ($n$ is mentioned in each figure legend). Non-parametric Mann–Whitney $U$ test (two-tailed) was used to assess significance levels and ANOVA was used to compare more than two groups. Statistical analysis was performed using GraphPad Prism version 6 software. For further statistical details, refer to each figure legend. For in vitro experiments, sample size required was not determined a priori. The experiments were not randomized.

**Reporting summary**. Further information on research design is available in the Nature Research Reporting Summary linked to this article.

## Data availability

The authors declare that all data supporting the findings of this study are available within the article and its supplementary information. The bulk RNAseq data generated in this study have been deposited in the GEO database under accession code: GSE109566. The single-cell RNA-Seq data have been deposited in the GEO database under accession code: GSE131909. Materials, other data, and Excel files with significantly differentially regulated genes and pathways are available upon reasonable request to the authors. A step-by-step protocol describing the establishment, maintenance, and differentiation of organoids can be found at Nature Protocol Exchange with DOI 10.21203/rs.2.12413/v1[49].

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

## Acknowledgements

We thank the members of the Epithelial Carcinogenesis Group and the CNIO Bioinformatics Unit for valuable discussions; N. del Pozo, Y. Gonzalo, and R. Serrano for help with animal experimentation; E. Carrillo-de-Santa-Pau and O. Graña for help with bioinformatics analyses; E. Osinaga for help with lectin assays; A. García-España for providing antibodies; A. Bigas, D. Chondronasiou, G. Mata Martínez, M. Deblas, J. Gómez-Alonso, J. González, A. Klinakis, S. Lowe, N. Malats, L. Martínez, J. L. Martínez-Torrecuadrada, M. Pérez Martínez, M. Ricote, M. Blasco, and M. Serrano for other valuable contributions; and J. Paramio, P. Martinelli, and M. Marqués for comments to the manuscript. We thank E. Batlle and his group for many valuable suggestions. This work was supported, in part, by grants from Spanish Ministry of Economy, Industry and Competitiveness (SAF2016-76377-R), CIBERONC (CB16/12/00453 and CB16/12/00273), and Fundación Científica de la Asociación Española Contra el Cáncer. CNIO is supported by Ministerio de Ciencia, Innovación y Universidades as a Centro de Excelencia Severo Ochoa SEV-2015–0510.

## Author contributions

Conceptualization: C.P.S., E.L., and F.X.R.; methodology: C.P.S., E.L., J.M.V., L.A.-E., A.F.-B., A.B., O.D., A.L., D.M., A.M., and .FX.R.; investigation: C.P.S., E.L., J.M.V., L.A.E., A.F.B., and A.B.; data curation: C.P.S., E.L., J.M.V., D.M., and F.X.R.; writing—original draft: C.P.S. and F.X.R.; writing—review & editing: C.P.S., E.L., J.M.V., A.F.B,. A.B., O.D., A.L., A.M., and FXR; funding acquisition: F,X,.R. and A.M.; resources: F.X.R.

## Competing interests

The authors declare no competing interests.
