## [Peer Review File · Nature Communications]

Reviewers' Comments:

Reviewer #1:

Remarks to the Author:

The authors describe the establishment and characterisation of normal mouse urothelial organoids (NMU-o) generated from a sub-population of CD49f+ urothelial basal "stem cells" sorted by flow cytometry. Organoids are maintained and propagated in Matrigel and require EGF/Wnt stimulation, whilst omission of growth factors induces differentiation-associated changes, particularly when PPAR γ is stimulated. The abstract refers to the powerful nature of the organoids for examining the genetic steps of bladder cancer, but this is not a focus of the current manuscript. The role of Notch mentioned in the title is not covered in the Abstract and seems a minor part of the overall study. While studies have been published which characterise the culture of organoids propagated from bladder cancer cells (Yoshida et al., 2018), this appears to be the first study to apply the techniques to create organoids with normal (mouse) urothelial cells. The methods section detailing the protocol used to establish NMU-o appear detailed enough to be reproduced. While the authors' claims of growing NMU-o with similar expression patterns to in vivo bladder is novel, the evidence provided in certain figures is inconclusive/insufficient. A large amount of the evidence provided is of RT-qPCR of gene targets where no significant difference between differentiation states is observed, or IF/IHC images that are too small and currently equivocal. Western blots of said targets using organoid lysates could help to strengthen the argument of differences in protein expression between organoids in different states/treatments (primarily Figures 2-4). Further evidence to support that the differentiated organoid represents an accurate in vitro model of mouse urothelium would be of interest to researchers working in the field of bladder biology. However, the authors do not attempt to rationalise the significance of bladder growth/differentiation as a direct consequence of the inclusion of EGF/WNT/TGF β agonists/antagonists in medium. The authors are encouraged to be more specific in the introduction and discussion when referring to urothelia from different species, as subtle but important differences of phenotype exist between mouse and human urothelia (for example, keratin 14 is absent from human urothelium but present in mouse urothelium and mouse expression of uroplakins is suprabasal rather than superficial-restricted as in humans).

Major points

1. Micrograph images generally throughout paper are far too small or at too low scale to be clear.
2. Cell input. The results section indicates that the urothelial scrapings used to isolate cells for organoid development contained only 12% epithelial cells based on EpCAM positivity, yet no cell selection was performed to establish organoids. In Figure 1A: the tissue localised-expression of CD49f protein is a key point, as these are the cells purported to display stem cell-like features and CD49f "selectively labels basal cells". The micrograph of CD49f expression in mouse urothelium is difficult to assess, but it appears that the antibody labels not only basal urothelial cells, but also cells of the stroma (as determined by placement of white dotted line for basement membrane). The text should explain that CD49f expression is not urothelium-restricted in the mouse bladder and discuss what this implies for the tissue derivation of the organoids.
3. It is perhaps surprising that organoids did not form any recoverable tissue when implanted under the kidney capsule. Although I would not expect tumours, I might have expected some normal tissue development and recovery, but these results are not shown.
4. Figure 1E: The result of organoid-forming capacity of CD49f^{high}WGA⁻ cells is not shown. Strong labelling of WGA is found in "the cytoplasm of umbrella cells...while the remaining urothelial cells are weakly labelled" yet CD49f^{high}WGA⁺ are the cells that are gated and used as the model "basal" population. It is not explained why selection of luminal cell-specific WGA expression would select a population of basal stem cells with the highest organoid-forming capacity. The FACS analysis used in Figure 1 could be improved as there does not appear to be good separation between positive and negative cell populations at any stage. Is this a problem with incomplete titration of antibodies?
5. Figure 2E-G: The functional competence of organoids is purported to be shown with FITC-dextran and FRAP in so-called "urothelial barrier assays", but what the assays actually measure has not been validated. The accompanying figures (and videos) are confusing as they suggest that

no change occurs between the two conditions post-bleaching. This therefore makes the downstream quantification of P vs D recovery hard to understand/interpret.

6. Figure 2I: Promising data but images of expression of more targets beyond KI67 would be good to establish the plasticity of the organoids after switch back to proliferative medium from differentiated medium. Expect to see switch back to KRT5+/TP63+/PPAR γ -/FOXA1- phenotype?

7. Figure 3: As with previous figure, IF of KRT14/KRT5 suggests no significant change in expression of proteins in proliferative/differentiated organoids, even following PPAR γ -driven differentiation by Rosiglitazone/Erlotinib (RZ/ER). PPAR γ is not visualised by IF as a control in Figure 3A to confirm that RZ/ER treatment was successful. Quantification of lumen-containing organoids has large error bars, while proliferative organoids + RZ/ER have a greater amount of lumen-containing organoids than differentiated organoids with the same treatment. This conflicts with the earlier figure that proposes differentiated organoids have "increased lumen formation". Authors use a heatmap to convey changes in gene expression (qRT-PCR) between proliferative/differentiated organoids treated with RZ/ER, when in previous (and upcoming) figures they have used graphs comparing conditions for the individual genes. Using this method of data visualisation gives no indication as to whether there is a significant difference in expression between treatments for any of the gene targets. Expression of Pparg and Foxa1 (a PPAR γ -driven target) is higher in differentiated organoids treated with PPAR γ antagonist T0070907 than in differentiated organoids after RZ/ER treatment, suggesting that PPAR γ blockade has not worked? By contrast, Figure 3C (heatmap) suggests that differentiated organoids treated with T0070907 regain expression of Cd44/Cd49f which is not discussed, but is interesting if reproducible.

8. Figure 4A-B: Expression of Cldn4 is higher in differentiated organoids while Cldn8 is higher in proliferative organoids. This is contrary to expression in normal mouse urothelium (Supp. Fig. 4E) which has low Cldn4 and high Cldn8 expression. Cldn3 is shown to have higher expression in differentiated organoids but is not included in the graph of Claudin expression in Supp. Fig. 4E for reasons not explained.

9. Figure 4D: Cldn4 was found to be upregulated in differentiated organoid samples. Text states that "selected mRNA expression changes were confirmed in independent samples" although expression of CLDN4 appears similar in proliferative and differentiated organoids by IHC labelling.

Minor points

1. Some references not correctly formatted (Gandhi et al. 2013).

2. Figure 1F-G: The optimum organoid forming capacity of the CD49f^{high}/CD44^{high} population is of limited relevance to the rest of the paper, and as such would be more suitable as part of a supplementary figure.

3. Figure 1H: Confusing/unclear layout for figure.

4. Figure 2A: Over-interpretation: "A dramatic morphological change was observed, including a reduced number of cell layers and an increase in lumen formation, indicative of barrier function." No quantification of number of cell layers, lumen dimensions or barrier function is performed in this figure.

5. Figure 2H: Although explained well in methods, diagram to explain experimental strategy of Fig. 2I is confusing; suggests that P cells are collected for histology at 7 days while D \rightarrow P cells are collected at 7 days later.

6. Supplementary Figure 4F: Images of bladder from the Human Protein Atlas are used to indicate localisations of CLDN1, CLDN3 and CLDN4. As human bladder samples are used, this may not coincide with the expected localisation of these proteins in mouse bladder.

7. Figure 5C: Changes in expression of Hey1 are shown as a primary target in this figure, but the importance of the gene (target gene of Notch signalling) is not described in the text.

8. Figure 5F: Although a trend is observed towards lower expression of differentiation-associated genes when treating organoids with a Notch inhibitor, Upk3a is the only target to have significant reduction in expression. "Notch inhibition... significantly reduced the expression of Upk1b, Upk2, Upk3a, and Krt20 mRNAs, and it completely abrogated lumen formation" is therefore an over-interpretation of the data.

9. Discussion, paragraph 4: "We speculate that... exposure to urine may additionally contribute to the formation of this cell type and the activation of its characteristic terminal differentiation

program." Stahlschmidt et al. have previously determined that lack of exposure to urine has no effect on the terminal differentiation of the urothelium (Stahlschmidt et al., 2005).

References

- Carpenter et al. *Kidney International*, 2016; 89(3), 612–624.
<https://doi.org/10.1016/j.kint.2015.11.017>
- Stahlschmidt et al. *Kidney International*, 2005; 68(3), 1032–1040.
<https://doi.org/10.1111/j.1523-1755.2005.00495.x>
- Yoshida et al. *Oncotarget*, 2018; 9(13), 11060–11070.
<https://doi.org/10.18632/oncotarget.24308>

Reviewer #2:

Remarks to the Author:

The manuscript from Santos et al. aims to establish the first organoid culture model for urothelial epithelium from mouse healthy urothelial cells. Using this approach, the authors also establish that a population of CD49^{high} cells are the starting population for these cultures and behave as stem cells in a dish and identify notch signaling as the pathway involved in the differentiation of these cells (at least in vitro)

Major comments:

- 1) To confirm that the organoids generated are indeed urothelial and recapitulate the original tissue histology H&E sections from the original mouse urethelial tissue are required. This comparison to original tissue architecture is missing in Suppl Fig1
- 2) The authors mention that CD49^f labels basal cells and these are the ones having organoid formation capacity. Alos a subpopulation double + for CD49^f CD44 has higher colony formation efficiency. This is interesting, but staining of these Cd44 and Cd49 cells in vivo is missing to help identifying these sub-populations in vivo. The co-labeling would help interpreting the subsequent sorting for double + Cd49 and Cd44. Further, it would be advisable to have co-labeling with another basal cell marker (e.g Krt5 or other).
- 3) Regarding the clonal experiments, it is unclear why these are performed from single sorted Cd44⁺ Cd49^{high} cells from organoids instead than from the tissue. Also, the GFP/Tomato clonal experiment is very difficult to understand, is unclear to me what the authors want to demonstrate by sorting single cells from organoids from Tom or GFP mice.
While the organoid clonal experiment is interesting, and allows concluding that the cells that maintain the organoids are the Cd49⁺ cells it would be more compelling to see that from the tissue. With an almost 50% colony formation efficiency (presented in Figure 1g) for the double + population, the authors should be able to perform clonal experiments directly from the tissue, and not from the organoids. That would inform us on the stem cell potential of this double positive population in vivo.
- 4) The authors show that the organoids after differentiation express some of the markers of the intermediate cells. This is very interesting. The RNA levels are very compelling but the UPK3A and PPAR γ stainings indicate that the real number of cells that differentiate is very little. Which is the real percentage of differentiation? How does this compare to the expression in the tissue?
- 5) The authors then go one step further and perform a functional assay to test the permeability and barrier function of the organoids. The results are interesting and indicate that the differentiated organoids indeed show better barrier function than the proliferating ones. I wonder how does this

compare to normal tissue. Could the authors compare that to the function of the tissue either in vivo or in explants? That would give us an indication whether the barrier function is similar to in vivo or not.

5) I do not understand the statement that urothelial organoids display plasticity. The experiment that the authors do here is to take their differentiated organoids and put them back into complete medium and they observe the re-appearance of Ki67+ cells after which they conclude that the cultures exhibit plasticity. My concern here is that the most likely (according to the expression marker) is that the cells obtained in DM medium are not terminally differentiated and hence do not undergo a fully de-differentiation program but instead they are in an intermediate state so, more than calling this plasticity the most likely is that they just re-enter cell cycle. Would not that be expected from a non-terminally differentiated cell? One could say they undergo plasticity if they are fully differentiated cells. Ki67 only is not a sufficient proof to say plasticity. I would suggest the authors to try the same experiment with their improved differentiation medium with the Rz and Egfr inhibitor

6) in the last section of the manuscript the authors perform genome-wide RNAseq analysis and some inhibitor tests to conclude that Wnt and Notch are involved in the self-renewal and differentiation of the urethelial epithelium. This is interesting. While not entirely novel, as canonical notch signaling has been described inactive in bladder cancer (see PMID: 25167871 and PMID: 25194568), this seems to be the first evidence that notch is involved in normal urethelial differentiation.

Overall, the manuscript is interesting, however bench-marking the findings with the in vivo results seems a clear shortcoming from the paper. Are these Cd49fhigh cells also stem cells in vivo? Is notch signaling also involved in the differentiation of these cells in vivo? Confirming that the results observed also occur physiologically is missing and would significantly strengthen and validate the drawn conclusions.

Minor comments:

Line 127: should read "these markers were"

Line128: "all experiments reported below were performed without cell sorting..." it is not clear what are the authors doing as in the following paragraph it says that EpCAM+ cells. Some clarification is needed . Did they sort or not?

Line 179 typo: displaying

Reviewer #3:

Remarks to the Author:

The current manuscript describes the development of an organoid culture based system for normal mouse urothelium. This is an innovative and significant study, as this is the first organoid system enabling the study of normal urothelial differentiation. The study is well conducted and generally well communicated. Major strengths of the work include clear evidence of cytodifferentiation in the organoid system, a clear protocol, and RNA-seq studies which should become publically available following publication. Weaknesses include an over reliance on pharmacologic approaches for the study of Wnt and Notch pathways, and their contribution to urothelial differentiation. That being said-this is an important study which will serve has an important resource for the community.

Minor concerns:

1. Page 3: "organoids are derived from stem/progenitor cells" Is this strictly true/known/confirmed? If these cells cannot expand in vivo, does this not potentially suggest they

are not from a stem cell population? Does it even matter? The point is that we don't have a model to study normal urothelial differentiation (other than NHU cells which have not been used in organoid cultures to my knowledge). This is particularly true more murine studies. Just curious about what the authors think...does it matter if the organoids are (or are not) derived from stem/progenitor cells?

2. Page 3: Citation of the Foxa1 KO paper (PMID 29507831) which shows squamous differentiation,, as well as the Gandhi Mendelsohn paper from 2013 (PMID 23993789) which showed the importance of retinoid signaling would seem pertinent here.
3. Figure 1C: why are the ratios so low for the basal markers? Is this just a reflection of the smaller basal cell number in the organoids? Just curious.
4. For figure 1 the conclusion is the CD49f population is the key organoid forming cell type. How was input cell viability confirmed following cell sorting?
5. What were the number of cells that were used for sub-cutaneous/renal graft implants? Did the input cells not grow or were they not able to be found following implantation?
6. Some of the figures in figure 1 are discussed out of order, making the reading a little difficult.
7. Figure 2C: the bottom panel (under the proliferative growth conditions) look rather squamous. This would seem to make sense as squamous differentiation is probably a default pathway in urothelium. Am curious about the expression of desmoplakin and desmoglein proteins under these conditions.
8. Some things were a bit confusing to me based on what was written on page 16 and data presented in Figure 5. CHIR99021 seems to be a GSK3 antagonist, so not a WNT agonist per se, correct? The authors state "In differentiating organoids, CHIR99021 induced marked morphological changes: organoids displayed a more solid aspect and a significant reeduction of lumen formation." First it looks like manipulation of Wnt and NOTCH signaling are presented in the figure out of order from the text (making the data hard to interpret). In panel B there appears to be some overlap between the data points following DAPT and IWP2 treatment. Also why would a rduction in lumen formation be interpreted as differentiation promoting? Seems difficult to interpret these findings without H&E or IF for markers like Krts, Pparg, Foxa1, Upks...etc.
9. What does NOTCH and nuclear beta cateinin look like in Figure 3?
10. Will the RNA-seq become publically available following publication?

Major concerns

1. Pharmacologic approaches can be informative, and point investigators in the right direction. However pharmacologic approaches are best used in conjunction with genetic gain of function/loss of function studies. Can the investigators knock-down/knock-out Wnt or Notch and test the impact on differentiation to substantiate the pharmacology studies? In reality the true strength of any organoid system lies in the ability of investigators to perform these types of studies.

Reviewers' comments:

Response to all reviewers

We appreciate the comments received from all 3 reviewers. We have made every effort to satisfy their requests and improve the manuscript. The genetic experiments carried out have not been satisfactory and further work to provide genetic evidence on the role of Wnt and Notch pathways in urothelial differentiation will take several months. However, the new data included in the revised manuscript clearly strengthen our conclusion that Notch pathway activation is involved in urothelial differentiation. Furthermore, we provide unique single-cell RNA-seq expression data supporting this notion and validating the potential of the organoids to address these questions. We believe that, together with the new cellular tools, this represents a very substantial contribution to the field.

Reviewer #1 (Remarks to the Author):

The authors describe the establishment and characterisation of normal mouse urothelial organoids (NMU-o) generated from a sub-population of CD49f+ urothelial basal "stem cells" sorted by flow cytometry. Organoids are maintained and propagated in Matrigel and require EGF/Wnt stimulation, whilst omission of growth factors induces differentiation-associated changes, particularly when PPAR γ is stimulated. The abstract refers to the powerful nature of the organoids for examining the genetic steps of bladder cancer, but this is not a focus of the current manuscript. The role of Notch mentioned in the title is not covered in the Abstract and seems a minor part of the overall study.

While studies have been published which characterise the culture of organoids propagated from bladder cancer cells (Yoshida et al., 2018), this appears to be the first study to apply the techniques to create organoids with normal (mouse) urothelial cells. The methods section detailing the protocol used to establish NMU-o appear detailed enough to be reproduced. While the authors' claims of growing NMU-o with similar expression patterns to in vivo bladder is novel, the evidence provided in certain figures is inconclusive/insufficient. A large amount of the evidence provided is of RT-qPCR of gene targets where no significant difference between differentiation states is observed, or IF/IHC images that are too small and currently equivocal. Western blots of said targets using organoid lysates could help to strengthen the argument of differences in protein expression between organoids in different states/treatments (primarily Figures 2-4).

Further evidence to support that the differentiated organoid represents an accurate in vitro model of mouse urothelium would be of interest to researchers working in the field of bladder biology.

However, the authors do not attempt to rationalise the significance of bladder growth/differentiation as a direct consequence of the inclusion of EGF/WNT/TGFB agonists/antagonists in medium. The authors are encouraged to be more specific in the introduction and discussion when referring to urothelia from different species, as subtle but important differences of phenotype exist between mouse and human urothelia (for example, keratin 14 is absent from human urothelium but present in mouse urothelium and mouse expression of uroplakins is suprabasal rather than superficial-restricted as in humans).

General response

1. We have now included western blots for selected proteins using new experiments performed to substantiate the findings (Figures 2 and 7).
2. We have strengthened the evidence that the normal urothelial organoids described provide unique opportunities to address the mechanisms involved in growth/differentiation despite the fact that "complete" urothelial differentiation is not achieved.
3. We have made a clear point in the Introduction/Discussion about the interspecies differences and about the possibility that there is not a perfect correlation between RNA and protein expression levels.

4. The additional experiments on gamma secretase inhibition and the single cell RNA-Seq data critically expand the relevance of our findings on the role of Notch in urothelial differentiation.
5. The paper by Yoshida et al. does not really report on organoids: they cultured bladder cancer cell lines in suspension, which is very different. Therefore, we have not included this reference as part of previous urothelial organoid work.

Major points

1. Micrograph images generally throughout paper are far too small or at too low scale to be clear. We thank the referee for this comment; we have gone through all the figures and have made efforts to improve the images.

2. Cell input. The results section indicates that the urothelial scrapings used to isolate cells for organoid development contained only 12% epithelial cells based on EpCAM positivity, yet no cell selection was performed to establish organoids. In Figure 1A: the tissue localised-expression of CD49f protein is a key point, as these are the cells purported to display stem cell-like features and CD49f “selectively labels basal cells”. The micrograph of CD49f expression in mouse urothelium is difficult to assess, but it appears that the antibody labels not only basal urothelial cells, but also cells of the stroma (as determined by placement of white dotted line for basement membrane). The text should explain that CD49f expression is not urothelium-restricted in the mouse bladder and discuss what this implies for the tissue derivation of the organoids.

We agree that this is an important finding. We have now added a sentence in the text to indicate that CD49f is not restricted to epithelial cells but is also expressed in some cells in the lamina propria (as is already known). Most importantly, we have added a new panel (Figure 1a) which clearly shows the double staining of CD49f and KRT5. This panel now shows that KRT5 has a broader distribution than CD49f and that cells displaying high CD49f expression are restricted to the basal layer of the urothelium. However, we do not rule out that cells expressing low level CD49f expression may occur in suprabasal cells.

Appropriate changes have been made in the text.

3. It is perhaps surprising that organoids did not form any recoverable tissue when implanted under the kidney capsule. Although I would not expect tumours, I might have expected some normal tissue development and recovery, but these results are not shown.

Indeed, we were surprised about this but - as indicated in the text - we used appropriate controls that grew, indicating that this was not due to a technical problem. However, to further examine this point, we have repeated the experiments by implanting cells under the kidney capsule, using other samples as controls, and have confirmed the lack of growth of the organoids in immunocompromised mice. The new experiments are reported in the Methods and Results sections. We do not feel that it is useful to provide these negative results in a more detailed manner but we are providing an image for the referee with the positive control used in these experiments.

4. Figure 1E: The result of organoid-forming capacity of CD49f^{high}WGA⁻ cells is not shown. Strong labelling of WGA is found in “the cytoplasm of umbrella cells...while the remaining urothelial cells are weakly labelled” yet CD49f^{high}WGA⁺ are the cells that are gated and used as the model “basal” population. It is not explained why selection of luminal cell-specific WGA expression would select a population of basal stem cells with the highest organoid-forming capacity. The FACS analysis used in Figure 1 could be improved as there does not appear to be good separation between positive and negative cell populations at any stage. Is this a problem with incomplete titration of antibodies?

We are sorry if the data was presented in a manner that was not clear. Using immunofluorescence, WGA-high cells are restricted to the luminal umbrella layer. Using FACS, WGA binding - at variable intensity - is found in all cells. Therefore, there is not - by FACS - a CD49f^{high}-WGA-negative population to test.

Therefore, basal cells are CD49f high-WGA-positive while suprabasal cells are CD49f low-WGA-positive; among the latter, "small" and "large" cell populations are found corresponding to the intermediate and umbrella cells, respectively. We have described this better in the text and have modified Fig. 1d accordingly.

Regarding the FACS analysis, WGA is a continuum and we resorted to size for separation. We do not think that this is a problem derived from the titration of the antibodies which were, indeed, titrated appropriately.

5. Figure 2E-G: The functional competence of organoids is purported to be shown with FITC-dextran and FRAP in so-called "urothelial barrier assays", but what the assays actually measure has not been validated. The accompanying figures (and videos) are confusing as they suggest that no change occurs between the two conditions post-bleaching. This therefore makes the downstream quantification of P vs D recovery hard to understand/interpret.

We modestly disagree with the referee. This type of assay has previously been used for intestinal organoids and we are the first to apply it to urothelial cells and organoids. We have made efforts to improve the presentation so that the results are clearer and we hope that this is now satisfactory to this reviewer; we have kept the general design of the figure used in the previous version of the manuscript given that the other two referees found this figure clear.

6. Figure 2I: Promising data but images of expression of more targets beyond KI67 would be good to establish the plasticity of the organoids after switch back to proliferative medium from differentiated medium. Expect to see switch back to KRT5+/TP63+/PPARγ-/FOXA1- phenotype?

Figure 2I is now Figure 5c. We appreciate the comment as we feel that this is an important experiment indicating that the differentiated organoids contain cells that are able to re-enter the cell cycle and display a basal-like phenotype. Following the referee's suggestion, we have now expanded the analysis to show expression of TP63 in the organoids switched to complete medium. We only provide immunostainings for TP63 because this marker is best at identifying the presence of basal-like cells. By contrast, KRT5, PPARγ and FOXA1 are more broadly expressed in basal and suprabasal/luminal cells in the organoids (see Figure 2g).

7. Figure 3: As with previous figure, IF of KRT14/KRT5 suggests no significant change in expression of proteins in proliferative/differentiated organoids, even following PPARγ-driven differentiation by Rosiglitazone/Erlotinib (RZ/ER). PPARγ is not visualised by IF as a control in Figure 3A to confirm that RZ/ER treatment was successful.

We agree with the referee. This may reflect the half-life of the keratins or the fact that differentiation is incomplete.

Quantification of lumen-containing organoids has large error bars, while proliferative organoids + RZ/ER have a greater amount of lumen-containing organoids than differentiated organoids with the same treatment. This conflicts with the earlier figure that proposes differentiated organoids have "increased lumen formation".

The large error bars result from the fact that there is wide heterogeneity in the organoid cultures (please see Figure 4b), a fact that is well acknowledged in the field. We think that there may be in some cases a reduction in epithelial impermeability in the RZ+Erlo condition due to a fraction of cells dying at the luminal side of the organoid.

Authors use a heatmap to convey changes in gene expression (qRT-PCR) between proliferative/differentiated organoids treated with RZ/ER, when in previous (and upcoming) figures they have used graphs comparing conditions for the individual genes. Using this method of data visualisation gives no indication as to whether there is a significant difference in expression between treatments for any of the gene targets.

We use both methods - bar graphs and heatmaps - with the aim of providing clear messages but - indeed - heat maps are less indicative of quantitative differences. To provide this detailed information, we have now added new graphs in Suppl. Figure 3a.

Expression of Pparg and Foxa1 (a PPAR γ -driven target) is higher in differentiated organoids treated with PPAR γ antagonist T0070907 than in differentiated organoids after RZ/ER treatment, suggesting that PPAR γ blockade has not worked? By contrast, Figure 3C (heatmap) suggests that differentiated organoids treated with T0070907 regain expression of Cd44/Cd49f which is not discussed, but is interesting if reproducible.

We believe that activation of PPAR γ targets, which would be affected by T0070907 and expression of PPAR γ transcripts itself are different and therefore we do not agree that this is indicative that PPAR γ blockade has not worked. Indeed, we tested the same drugs on RAW 264.7 cells demonstrating their activity (reported in Methods section).

Regarding the changes in Cd44 and Cd49f in the presence of T0070907, we agree that this is an interesting and have added a comment in the main text, as suggested by the referee.

8. Figure 4A-B: Expression of Cldn4 is higher in differentiated organoids while Cldn8 is higher in proliferative organoids. This is contrary to expression in normal mouse urothelium (Supp. Fig. 4E) which has low Cldn4 and high Cldn8 expression.

We believe that it is difficult to compare because the proportion of differentiated cells in the mouse urothelium is relatively low (these cells are larger than the basal cells). We do not think that there is necessarily a contradictory finding here.

Cldn3 is shown to have higher expression in differentiated organoids but is not included in the graph of Claudin expression in Supp. Fig. 4E for reasons not explained.

Claudin 3 was not detected in the urothelial sample, this may be related to gene expression or to the efficiency of the PCR. We do not think that there is necessarily a contradiction here, either.

9. Figure 4D: Cldn4 was found to be upregulated in differentiated organoid samples. Text states that "selected mRNA expression changes were confirmed in independent samples" although expression of CLDN4 appears similar in proliferative and differentiated organoids by IHC labelling. We agree with the referee that there appears to be some inconsistency here but we have improved the background of the microphotograph in Figure 6c and we do think that expression of CLDN4 is higher (at the protein level) in the differentiated organoids.

Minor points

1. Some references not correctly formatted (Gandhi et al. 2013).

Thanks for picking up these inconsistencies; we have now carefully reviewed all references for adequate format.

2. Figure 1F-G: The optimum organoid forming capacity of the CD49f^{high}/CD44^{high} population is of limited relevance to the rest of the paper, and as such would be more suitable as part of a supplementary figure.

We agree that this is not critical to our manuscript. However, CD44 has been proposed to be a stem cell marker in other systems and we feel that this may be valuable information that fits in this figure.

3. Figure 1H: Confusing/unclear layout for figure.

We appreciate the comment that the figure needs clarification and we have made our best to provide a better legend to this panel.

4. Figure 2A: Over-interpretation: "A dramatic morphological change was observed, including a reduced number of cell layers and an increase in lumen formation, indicative of barrier function."

No quantification of number of cell layers, lumen dimensions or barrier function is performed in this figure.

We appreciate this comment and have now performed a detailed quantitative analysis of the features of the organoids in both conditions. Specifically, we have measured organoid external diameter, organoid lumen diameter, and layer thickness and have performed rigorous statistical analysis in order to support our statements.

5. Figure 2H: Although explained well in methods, diagram to explain experimental strategy of Fig. 2I is confusing; suggests that P cells are collected for histology at 7 days while D δ P cells are collected at 7 days later.

We have now revised this figure which was, indeed, confusing. We have now modified and hope that it is clear. Thanks for making the point.

6. Supplementary Figure 4F: Images of bladder from the Human Protein Atlas are used to indicate localisations of CLDN₁, CLDN₃ and CLDN₄. As human bladder samples are used, this may not coincide with the expected localisation of these proteins in mouse bladder.

We agree with the referee's comment and have made a point about this in the main text of the manuscript. As requested, we have made a point in the Introduction that there are substantial species-related differences in the organization of the urothelium as well as in the Discussion.

7. Figure 5C: Changes in expression of Hey₁ are shown as a primary target in this figure, but the importance of the gene (target gene of Notch signalling) is not described in the text.

We have included Hey₁, as well as Hes₁, because it is a Notch pathway target gene and effector. However, the expression levels of Hey₁ are much lower than those of Hes₁ as shown both by RT-qPCR (approximately 4 cycles higher for Hey₁ although this may be affected by PCR efficiency) and by the single cell analyses.

8. Figure 5F: Although a trend is observed towards lower expression of differentiation-associated genes when treating organoids with a Notch inhibitor, Upk_{3a} is the only target to have significant reduction in expression. "Notch inhibition... significantly reduced the expression of Upk_{1b}, Upk₂, Upk_{3a}, and Krt₂₀ mRNAs, and it completely abrogated lumen formation" is therefore an over-interpretation of the data.

We agree that the previous version of the manuscript provided a relatively soft proof of the effect of Notch inhibition on differentiation.

Unfortunately, despite major efforts, we have not been able to provide additional evidence on the role of Notch using genetic tools. Specifically, we have attempted:

- to generate mice with inducible expression of Notch1C in the urothelium using Ub-CreERT. We have screened more than 100 embryos for the inducible Notch1C allele and the Ub-CreERT allele and none of them were positive for both alleles, suggesting early embryo lethality due to leaky Cre activity.

- to infect urothelial organoids with lenti-sh targeting Notch (for loss-of-function experiments) and with retroviruses carrying a Notch1C cDNA (for gain-of-function experiments). Unfortunately, these experiments have not been productive.

However, we now provide strong evidence of the role of Notch in normal urothelial differentiation, as shown in Figure 7. In this revised version we have:

- 1) used DBZ, a more potent, specific, gamma-secretase inhibitor as shown by the reduced expression of HES₁ at the protein level (Figure 7d). In addition, treatment with DBZ results in a highly significant reduction in lumen formation, expression of Upk_{1b}, Upk₂, and Upk₃ mRNAs, and expression of UPK_{1b} at the protein level. In contrast, TP6₃ is up-regulated as shown by both western blotting and immunohistochemistry. These results convincingly show - using two different gamma secretase inhibitors - that Notch inhibition is associated with inhibition of urothelial differentiation.

2) the single cell RNA-Seq data clearly show that Hes1 is selectively upregulated in cells with intermediate urothelial differentiation, supporting and refining the previous findings using bulk RNA-Seq.

g. Discussion, paragraph 4: "We speculate that... exposure to urine may additionally contribute to the formation of this cell type and the activation of its characteristic terminal differentiation program." Stahlschmidt et al. have previously determined that lack of exposure to urine has no effect on the terminal differentiation of the urothelium (Stahlschmidt et al., 2005).

Thanks for this information, which we were not aware of. We have added the reference to the manuscript. However, we have let our comment since it is possible that exposure urine is not required once the full differentiation program has been executed.

References

Carpenter et al. Kidney International, 2016; 89(3), 612–624. <https://doi.org/10.1016/j.kint.2015.11.017>

Stahlschmidt et al. Kidney International, 2005; 68(3), 1032–1040. <https://doi.org/10.1111/j.1523-1755.2005.00495.x>

Yoshida et al. Oncotarget, 2018; 9(13), 11060–11070. <https://doi.org/10.18632/oncotarget.24308>

Thanks for suggesting these citations. The reference by Stahlschmidt et al. has been added to the manuscript. As indicated above, we do not think that Yoshida et al. is pertinent to organoids as properly defined. We have not found a good justification to add the reference of Carpenter et al.

Reviewer #2 (Remarks to the Author):

The manuscript from Santos et al. aims to establish the first organoid culture model for urothelial epithelium from mouse healthy urothelial cells. Using this approach, the authors also establish that a population of CD49^{high} cells are the starting population for these cultures and behave as stem cells in a dish and identify notch signaling as the pathway involved in the differentiation of these cells (at least in vitro)

Major comments:

1) To confirm that the organoids generated are indeed urothelial and recapitulate the original tissue histology H&E sections from the original mouse urethelial tissue are required. This comparison to original tissue architecture is missing in Suppl Fig1

We have now added an H&E microphotograph in Figure 1A, as requested.

2) The authors mention that CD49 labels basal cells and these are the ones having organoid formation capacity. Alos a subpopulation double + for CD49 CD44 has higher colony formation efficiency. This is interesting, but staining of these Cd44 and Cd49 cells in vivo is missing to help identifying these sub-populations in vivo. The co-labeling would help interpreting the subsequent sorting for double + Cd49 and Cd44. Further, it would be advisable to have co-labeling with another basal cell marker (e.g Krt5 or other).

Thanks for these important suggestions, overlapping with those made by referee 1. We agree that this is an important finding. We have now added a new panel (Figure 1a) which clearly shows the double staining of CD49f and KRT5. This panel now shows that KRT5 has a broader distribution than CD49f and that cells displaying high CD49f expression are restricted to the basal layer of the urothelium. However, we do not rule out that cells expressing low level CD49f expression may occur in suprabasal cells. Appropriate changes have been made in the text.

Regarding CD44, there is a slight gradient of expression when using immunohistochemistry but the distinction of CD44-high vs. -low is much clearer using FACS (Figure 1e). That is the reason why we did not show these data but we are providing the referee with a microphotograph for consideration.

3) Regarding the clonal experiments, it is unclear why these are performed from single sorted Cd44+ Cd49high cells from organoids instead than from the tissue. We apologize that the results were not presented in a clear manner but - indeed - the single sorting experiments for organoid formation were performed using freshly isolated cells. We have now clarified this important point in the text and in the figure.

Also, the GFP/Tomato clonal experiment is very difficult to understand, is unclear to me what the authors want to demonstrate by sorting single cells from organoids from Tom or GFP mice. We apologize if the aim of this experiment was not clear; the other two referees did not seem to have problems. We believe that this is a valuable experiment since it shows that - starting from single cells derived from organoids - the new organoids are largely monoclonal since they are monocolour.

4) The authors show that the organoids after differentiation express some of the markers of the intermediate cells. This is very interesting. The RNA levels are very compelling but the UPK3A and PPAR γ stainings indicate that the real number of cells that differentiate is very little. Which is the real percentage of differentiation? How does this compare to the expression in the tissue? We agree that the expression of UPK3A and PPAR γ is patchy in the organoids whereas it is more extensive in the suprabasal cells in normal urothelium. However, we now provide information on the proportion of cells expressing uroplakins derived from the single cell RNA-Seq experiment (and this is clearly an underestimate given the low sensitivity of this technology at the current stage). These findings are now shown in Supplementary Figure 9.

5) The authors then go one step further and perform a functional assay to test the permeability and barrier function of the organoids. The results are interesting and indicate that the differentiated organoids indeed show better barrier function than the proliferating ones. I wonder how does this compare to normal tissue. Could the authors compare that to the function of the tissue either in vivo or in explants? That would give us an indication whether the barrier function is similar to in vivo or not.

We thank the referee for appreciating the value of the permeability test which, to the best of our knowledge has not been previously reported using normal urothelial cells. We have significantly strengthened these experiments by performing a detailed quantitative analysis of the features of the organoids. Specifically, we have measured organoid external diameter, organoid lumen diameter, and layer thickness and have performed rigorous statistical analysis in order to support our statements.

Unfortunately, we don't know how to compare this to barrier function in vivo and we are not aware of similar experiments ever having been performed in a manner that would be quantitatively comparable.

5) I do not understand the statement that urothelial organoids display plasticity. The experiment that the authors do here is to take their differentiated organoids and put them back into complete medium and they observe the re-appearance of Ki67+ cells after which they conclude that the cultures exhibit plasticity. My concern here is that the most likely (according to the expression marker) is that the cells obtained in DM medium are not terminally differentiated and hence do not undergo a fully de-differentiation program but instead they are in an intermediate state so, more than calling this plasticity the most likely is that they just re-enter cell cycle. Would not that be expected from a non-terminally differentiated cell? One could say they undergo plasticity if they are fully differentiated cells. Ki67 only is not a sufficient proof to say plasticity. I would

suggest the authors to try the same experiment with their improved differentiation medium with the Rz and Egfr inhibitor

Thanks for making this point. We agree with the referee that: 1) cells in the differentiated organoids are not terminally differentiated, 2) the findings reflect re-entry into the cell cycle, and 3) as shown by new experiments and Figure 4c, the switch to a more basal phenotype is also very clearly reflected in up-regulation of TP63. To make these points clearer, we have revised the text and replaced the term "plasticity" with "cell cycle re-entry".

6) in the last section of the manuscript the authors perform genome-wide RNAseq analysis and some inhibitor tests to conclude that Wnt and Notch are involved in the self-renewal and differentiation of the urethelial epithelium. This is interesting. While not entirely novel, as canonical notch signaling has been described inactive in bladder cancer (see PMID: 25167871 and PMID: 25194568), this seems to be the first evidence that notch is involved in normal urethelial differentiation.

As discussed in the paper, and also indicated by this referee, the issue of the role of Notch signaling in bladder cancer is not new (in fact, we are co-authors of a paper on this subject PMID 25574842). However, what is entirely new is the finding that Notch activation is required for normal urothelial differentiation. In fact, this is even more interesting considering that tumors with Notch inactivation, as we described earlier in PMID 25574842, display a less Urothelial and more Basal-Squamous phenotype.

Regarding Wnt, the group of P. Beachy - most notably - has shown a role of Wnt in urothelial regeneration. However, these studies have largely proposed that mesenchymal cells are the source of Wnt. Now, we unequivocally show - using the organoid bulk and single cell RNA-Seq data - that Wnt signals are also produced by epithelial cells and that Wnt ligands are largely restricted to basal cells with down-regulation occurring in more differentiated cells. This is entirely novel.

Overall, the manuscript is interesting, however bench-marking the findings with the in vivo results seems a clear shortcoming from the paper. Are these Cd49fhigh cells also stem cells in vivo? Is notch signaling also involved in the differentiation of these cells in vivo? Confirming that the results observed also occur physiologically is missing and would significantly strengthen and validate the drawn conclusions.

We understand this comment and agree that additional in vivo studies will be important. As indicated above, some of our planned experiments have not been possible due to the lack of embryos of the appropriate phenotype. Other mouse strains might help answering this question but acquiring these strains and performing crosses and experiments would take several months. We believe that we now provide significant new input into the relevance of our findings with the new studies using DBZ (which provide much clearer results than previous studies using DAPT) and with the single cell RNA-Seq of the organoid populations which clearly support the notion that Notch activation occurs transiently during differentiation.

Minor comments:

Line 127: should read "these markers were"

Thanks for picking up this error which has now been corrected.

Line128: "all experiments reported below were performed without cell sorting..." it is not clear what are the authors doing as in the following paragraph it says that EpCAM+ cells. Some clarification is needed. Did they sort or not?

Thanks for making this point; we have now appropriately made clear in the text which

experiments were performed with urothelial digests vs. organoid-derived cells as well as when EpCAM+ cells with sorting were used.

Line 179 typo: displaying

Thanks for picking up this typo which has now been corrected.

--

Reviewer #3 (Remarks to the Author):

The current manuscript describes the development of an organoid culture based system for normal mouse urothelium. This is an innovative and significant study, as this is the first organoid system enabling the study of normal urothelial differentiation. The study is well conducted and generally well communicated. Major strengths of the work include clear evidence of cytodifferentiation in the organoid system, a clear protocol, and RNA-seq studies which should become publically available following publication. Weaknesses include an over reliance on pharmacologic approaches for the study of Wnt and Notch pathways, and their contribution to urothelial differentiation. That being said-this is an important study which will serve has an important resource for the community.

Minor concerns:

1. Page 3: "organoids are derived from stem/progenitor cells" Is this strictly true/known/confirmed? If these cells cannot expand in vivo, does this not potentially suggest they are not from a stem cell population? Does it even matter? The point is that we don't have a model to study normal urothelial differentiation (other than NHU cells which have not been used in organoid cultures to my knowledge). This is particularly true more murine studies. Just curious about what the authors thing...does it matter if the organoids are (or are not) derived from stem/progenitor cells?

We agree with the referee, one major contribution of the work presented here is that it provides tools to consistently study and manipulate normal urothelial cells that can activate a tissue-specific differentiation program. To the best of our knowledge, there is not conclusive evidence that urothelial cells can activate multiple differentiation programs, therefore the discussion stem vs. progenitor may indeed be semantic and - as the referee - we do not wish to make a big point out of this.

2. Page 3: Citation of the Foxa1 KO paper (PMID 29507831) which shows squamous differentiation,, as well as the Gandhi Mendelsohn paper from 2013 (PMID 23993789) which showed the importance of retinoid signaling would seem pertinent here.

Thanks for making these suggestions; we have added the reference to these two important manuscripts to the paper.

3. Figure 1C: why are the ratios so low for the basal markers? Is this just a reflection of the smaller basal cell number in the organoids? Just curious.

Indeed, we think that it is the opposite interpretation (see that the ratio is Cd49f low/high). Just as a clarification, these data refer to primary dissociated cells. In any case, we take the point of the referee that this may be confusing and have changed the way the data are presented and moved this panel to Supplementary Figure 1f.

4. For figure 1 the conclusion is the CD49f population is the key organoid forming cell type. How was input cell viability confirmed following cell sorting?

This is an important point; we have now added a comment on cell viability in the Materials and Methods section (page 17) of the manuscript. Our data clearly show that the results are not due to differences in cell viability, which was similar in all three cell populations in two independent experiments:

Expt. 1

Epcam+ Cd49fhigh Cd44high 87%
Epcam+ Cd49fhigh Cd44low 80%
Epcam+ Cd49flow Cd44low 81%

Expt. 2

Epcam+ Cd49fhigh Cd44high 60%
Epcam+ Cd49fhigh Cd44low 50%
Epcam+ Cd49flow Cd44low 70%

5. What were the number of cells that were used for sub-cutaneous/renal graft implants? Did the input cells not grow or were they not able to be found following implantation?

The detailed information is provided in the Methods section. Mice were injected with 10^6 cells in 100 μ L Matrigel under the flank or 5×10^4 cells in 50 μ L Matrigel under the kidney capsule. A higher number of cells could not be administered in the kidney capsule due to technical constraints. We have repeated the experiments by implanting cells under the kidney capsule, using other samples as controls, and have confirmed the lack of growth of the organoids in immunocompromised mice. In this new experiment, we recovered samples 1, 2 and 4 weeks after injection and - after extensive sectioning - we could find any viable cells over this time period.

6. Some of the figures in figure 1 are discussed out of order, making the reading a little difficult.

Thanks for making this point. We have revised the figure and text to facilitate reading.

7. Figure 2C: the bottom panel (under the proliferative growth conditions) look rather squamous. This would seem to make sense as squamous differentiation is probably a default pathway in urothelium. Am curious about the expression of desmoplakin and desmoglein proteins under these conditions.

We agree with the referee that squamous differentiation is likely the default pathway in the urothelium and have made a point about this in the Discussion. While we do not feel that it is worthwhile including this in the paper, we are attaching analysis of the expression of desmoplakin and plakoglobin in normal urothelium and in the organoids for the referee's information.

8. Some things were a bit confusing to me based on what was written on page 16 and data presented in Figure 5. CHIR99021 seems to be a GSK3 antagonist, so not a WNT agonist per se, correct?

Correct, thanks. We have included this information in the text.

The authors state "In differentiating organoids, CHIR99021 induced marked morphological changes: organoids displayed a more solid aspect and a significant reduction of lumen formation." First it looks like manipulation of Wnt and NOTCH signaling are presented in the figure out of order from the text (making the data hard to interpret).

We have now completely changed the way these data are presented and we hope that this facilitates getting the message across to the readers.

In panel B there appears to be some overlap between the data points following DAPT and IWP2 treatment. Also why would a reduction in lumen formation be interpreted as differentiation promoting?

One important point is that while CHIR would affect Wnt signaling resulting from both exogenously added Wnt ligands and from endogenous Wnt ligands, IWP2 only affects endogenous ligands. We agree with the referee that this was confusing and have now made it clearer both in text and figures.

Seems difficult to interpret these findings without H&E or IF for markers like Krt5, Pparg, Foxa1, Upps...etc.

We agree and we have reduced the emphasis placed on Wnt in the manuscript. Accordingly, the Wnt-related figures have been moved to the Supplementary Materials and we have expanded the experiments with Notch (see below) instead as requested by all the referees.

9. What does NOTCH and nuclear beta catenin look like in Figure 3?

We tried to determine whether there are changes in Notch distribution in proliferative vs. differentiated organoids but - with the antibodies we used - we did not find such evidence. That is the reason why these results have not been included in the manuscript.

Regarding beta-catenin, we looked carefully for the presence of nuclear beta-catenin in normal urothelium and in the organoids and could not find any evidence for it. Only cytoplasmic staining was observed in the samples tested.

10. Will the RNA-seq become publicly available following publication?

Of course!!! All the data, the bulk RNA-Seq and the single-cell RNA-Seq included in the new version of the manuscript will be made publicly available to the community. In addition, we are happy to provide any relevant reagents/cells to anybody who needs them. This has always been our laboratory policy.

Major concerns

1. Pharmacologic approaches can be informative, and point investigators in the right direction. However pharmacologic approaches are best used in conjunction with genetic gain of function/loss of function studies. Can the investigators knock-down/knock-out Wnt or Notch and test the impact on differentiation to substantiate the pharmacology studies? In reality the true strength of any organoid system lies in the ability of investigators to perform these types of studies.

We completely agree with the referee and we have made major efforts to address these issues using genetic tools. However, we realize that it will take much more time to address these issues genetically and we feel that by strengthening our evidence on the role of Notch both with additional stronger gamma-secretase inhibitors and the single-cell RNA-Seq analysis we provide sufficiently novel and strong evidence to reach conclusions. We hope that both referees and editors will concur with this.

Reviewers' Comments:

Reviewer #1:

Remarks to the Author:

This is a revised manuscript. In revision the manuscript text has been lengthened with extensive description/characterisation of the organoids, but the manuscript rather fails to bring clear new specific insight or understanding of urothelial biology – such as the nature of the niche that maintains the urothelium. Given that mouse organoids have already been described in the literature (eg Mullenders et al 2019, from the Clevers' lab) and hence this work lacks some of the novelty it might once have had, I feel the manuscript would benefit from presentation in a more concise form. Although the authors do refer to the Mullenders publication, it is primarily to dismiss it as missing the opportunity to study urothelial biology and there is no examination of similarities/differences of technique.

Major points:

It is unclear throughout the manuscript if and how many mice bladders were pooled to establish organoids. It is equally unclear if and how many times the development of organoids and the characterisation experiments have been replicated or validated on independent organoid series. P4 line 137. The authors indicate that organoids are maintained in continuous culture for >1 year and passaged routinely. However, it is not clear how frequently they are passaged or how this relates to the statement that experiments reported relate to organoids at passage <10. How many passages are organoids after 1 year and do they show the same phenotype?

P4 line 126. This details the pre-organoid starting population of cells, where the epithelial cell component is only ~12% of the total. Although the authors indicate that organoids could be constituted using either the unselected population or from EpCAM-selected cells, it is not clear which approach was used for the majority of work reported. The histology of organoids indicates there is a non-epithelial compartment, but the organoid "stroma" is not characterised even though it presumably is responsible for supporting the epithelial compartment. Does this stromal cell compartment form a component in the downstream transcriptomic (and single cell?) analyses. The potential for the stroma to be contributing to the epithelial signalling axis seems incomplete. Other points.

The whole manuscript needs checking for grammar and minor errors of English

P3 line 78. Most urothelial pathologists would disagree that urothelium (a transitional epithelium) resembles skin (a stratified squamous epithelium) !

P3 line 80. Change sentence to start: "In mouse, basal cells....."

P5 line 158. If as stated, WGA labels a cytoplasmic component, how was it used in FACS sorting?

P6 line 214. The authors interpret the failure of T007907 to reverse differentiation as meaning that other differentiation pathways are involved. However, the authors should consider that as PPAR γ activation establishes the differentiated phenotype but is not required for maintenance, adding T007907 to already established differentiated urothelium would not be expected to act to reverse the phenotype.

P11 line 387. Reference included but has not been read by the authors as it does not support urine providing a differentiation-promoting signal (the opposite!).

P11 line 397. Relates to "cells with self-propagating capacity" when reference is to TERT-immortalised cells.

P12 line 421. Incorrect reference.

I am unclear why human bladder cancer cell lines are included in the Methods.

Reviewer #2:

Remarks to the Author:

This manuscript from Santos et al is indeed improved. The new additions , specially the scRna seq data significantly help in better characterizing the populations obtained in the organoids and majority of my concerns are well addressed.

My major concern, the benchmarking to in vivo tissue is half addressed, though, but I understand that the authors might have difficulties in accessing this material. In light of this, the authors should discuss these points clearer in the text, specially the barrier function assay, to not provide the readers with false expectations.

My only remaining points are listed below.

1-do not understand the comparison between the histologies of the tissue and organoids. Also the fact that they are separated in 2 different figures (main ad suppl) make difficult the comparison, specially for non experts. I would suggest placing them in the same figure for a side by side comparison.

2-Barrier function: this rev understands that quantifying barrier function in vivo might be difficult. In that case, then the claims in that respect should be toned-down since with the data presented it is impossible to tell whether the results are physiologically relevant and whether these organoids would prevent urine leakage as the in vivo tissue. This could be discussed in the text to make it clear to readers and not overestimate the technology.

Adressing these 2 last points on overstatements (by textual arrangements in text and discussion) are important as the organoid field is getting filled with papers of poor quality that claim results that are not supported by the data, which it only harms the field and organoid community.

Reviewer #3:

Remarks to the Author:

The authors have satisfactorily addressed my concerns.

REVIEWERS' COMMENTS:

Reviewer #1 (Remarks to the Author):

This is a revised manuscript. In revision the manuscript text has been lengthened with extensive description/characterisation of the organoids, but the manuscript rather fails to bring clear new specific insight or understanding of urothelial biology – such as the nature of the niche that maintains the urothelium. Given that mouse organoids have already been described in the literature (eg Mullenders et al 2019, from the Clevers' lab) and hence this work lacks some of the novelty it might once have had, I feel the manuscript would benefit from presentation in a more concise form. Although the authors do refer to the Mullenders publication, it is primarily to dismiss it as missing the opportunity to study urothelial biology and there is no examination of similarities/differences of technique.

We thank the referee for acknowledging the changes introduced in the revised version. We hope that the reviewer has appreciated the fact that our report provides a much more detailed account of the urothelial differentiation of the organoids than other published papers, even without consideration of the new data from the single cell RNA-sequencing experiments. A direct comparison with the Mullenders publication is not possible but we have made a point in the Discussion as to similarities and differences of technique and findings (page 10), as requested. We agree that this is a valuable and useful addition.

Major points:

It is unclear throughout the manuscript if and how many mice bladders were pooled to establish organoids. It is equally unclear if and how many times the development of organoids and the characterisation experiments have been replicated or validated on independent organoid series.

Following the suggestion of the Editor, we have deposited a step-by-step protocol in Nature Protocols Exchange so that all investigators can replicate our findings. The information regarding several of the points raised by the reviewer were already provided in the manuscript (page 15, Materials and Methods) but we are glad to provide a response. Specifically, we do not need to pool several bladders to establish organoids; bladder pooling was only performed when using FACS-sorted cell populations in order to enrich for the less abundant subsets of cells. Regarding replication or validation:

- we have used more than 10 independent organoid series
- unless stated otherwise, all important experiments were performed at least on 2 different organoid series

P4 line 137. The authors indicate that organoids are maintained in continuous culture for >1 year and passaged routinely. However, it is not clear how frequently they are passaged or how this relates to the statement that experiments reported relate to organoids at passage <10. How many passages are organoids after 1 year and do they show the same phenotype?

Following the suggestion of the Editor, we have deposited a step-by-step protocol in Nature Protocols Exchange so that all investigators can replicate our findings. Specifically:

- we have used 2 organoid series for the "young vs. old" comparison, at passages 4, 8, 12, and 22. We observe increased basal features in "old" organoids, as was already shown in Supplementary Figure 1 of the prior version of the manuscript
- all organoid cultures were split weekly, regardless of passage
- unless stated otherwise, all experiments reported in the paper are performed with organoids at passage <10, as indicated in the Methods section

P4 line 126. This details the pre-organoid starting population of cells, where the epithelial cell component is only ~12% of the total. Although the authors indicate that organoids could be constituted using either the unselected population or from EpCAM-selected cells, it is not clear which approach was used for the majority of work reported.

Following the suggestion of the Editor, we have deposited a step-by-step protocol in Nature Protocols Exchange so that all investigators can replicate our findings.

Specifically:

- as already described in the previous version of the manuscript: "All experiments reported were performed without cell sorting based on EpCam expression, using low-passage cultures (<10), unless stated otherwise". We do hope that this is clear now.

The histology of organoids indicates there is a non-epithelial compartment, but the organoid "stroma" is not characterised even though it presumably is responsible for supporting the epithelial compartment. Does this stromal cell compartment form a component in the downstream transcriptomic (and single cell?) analyses. The potential for the stroma to be contributing to the epithelial signalling axis seems incomplete.

We respectfully - but strongly - disagree with the referee on this point - which by the way was not raised in the prior round of review. It is correct that the urothelial digests contain a large majority of non-epithelial, as indicated in page 4 of the manuscript. However, the reviewer's statements on the occurrence of stromal cells in the organoids are simply not right. Our organoids do not contain a stromal component, as is very clearly shown from the morphology and from the single-cell analysis. Therefore, unlike suggested by the referee, the statement "The potential for the stroma to be contributing to the epithelial signalling axis seems incomplete." is out of place.

Other points.

The whole manuscript needs checking for grammar and minor errors of English

We thank the referee for this request. We have carefully reviewed the paper for grammar and English.

P3 line 78. Most urothelial pathologists would disagree that urothelium (a transitional epithelium) resembles skin (a stratified squamous epithelium) !

We have modified the sentence according to this comment in page 3.

P3 line 80. Change sentence to start: "In mouse, basal cells....."

We have modified the sentence, as requested.

P5 line 158. If as stated, WGA labels a cytoplasmic component, how was it used in FACS sorting?

WGA labels glycoproteins, which are found both in intracellular compartments and in the plasma membrane, thus allowing for sorting upon cell surface labeling. We appreciate this comment and have modified the sentence accordingly in page 5.

P6 line 214. The authors interpret the failure of T007907 to reverse differentiation as meaning that other differentiation pathways are involved. However, the authors should

consider that as PPAR γ activation establishes the differentiated phenotype but is not required for maintenance, adding T007907 to already established differentiated urothelium would not be expected to act to reverse the phenotype.

We understand the concerns of the referee but, sadly, it seems that she/he may have overlooked that T007907 was added to the organoids prior to inducing differentiation. Therefore, this point is not truly relevant. In addition, we do not think that there is enough evidence to rule out that PPAR γ is not required for the maintenance of the differentiated phenotype from a careful analysis of the literature. However, we have been cautious and have modified our statement in page 6.

P11 line 387. Reference included but has not been read by the authors as it does not support urine providing a differentiation-promoting signal (the opposite!).

Of course, we read the reference and we know that the conclusion from that paper is the opposite - which does not mean necessarily that urine could not have an effect in our system. The results of this experiment cannot be predicted. We have made a point in page 11 that our statement refers to the *in vitro* studies.

P11 line 397. Relates to “cells with self-propagating capacity” when reference is to TERT-immortalised cells.

We appreciate the comment and have modified the sentence accordingly in page 11.

P12 line 421. Incorrect reference.

Thanks for picking up this error; we have now replaced with references 22 and 23.

I am unclear why human bladder cancer cell lines are included in the Methods.

We imagine that the reviewer has examined Figures 2f and 7c, where the human bladder cancer cells have been included as controls for the western blot experiments.

--

Reviewer #2 (Remarks to the Author):

This manuscript from Santos et al is indeed improved. The new additions, specially the scRNA seq data significantly help in better characterizing the populations obtained in the organoids and majority of my concerns are well addressed.

We are glad that the referee considers that the manuscript is improved.

My major concern, the benchmarking to *in vivo* tissue is half addressed, though, but I understand that the authors might have difficulties in accessing this material. In light of this, the authors should discuss these points clearer in the text, specially the barrier function assay, to not provide the readers with false expectations.

My only remaining points are listed below.

1-do not understand the comparison between the histologies of the tissue and organoids. Also the fact that they are separated in 2 different figures (main and suppl) make difficult the comparison, specially for non experts. I would suggest placing them in the same figure for a side by side comparison.

Following the reviewer's requests, we have included now microphotographs of haematoxylin-eosin sections in Figure 2g, as requested to facilitate direct comparison.

2-Barrier function: this reviewer understands that quantifying barrier function *in vivo* might be difficult. In that case, then the claims in that respect should be toned-down since with the data presented it is impossible to tell whether the results are physiologically

relevant and whether these organoids would prevent urine leakage as the in vivo tissue. This could be discussed in the text to make it clear to readers and not overestimate the technology.

We appreciate this comment and have toned-down our statements and have made a specific point in the Discussion, stating that we cannot compare our results with in vivo barrier function (pages 6 and 11).

Addressing these 2 last points on overstatements (by textual arrangements in text and discussion) are important as the organoid field is getting filled with papers of poor quality that claim results that are not supported by the data, which it only harms the field and organoid community.

We completely agree with the referee that it is important to be cautious and we hope that our paper represents an addition to the "high-quality" papers in the field, contributing to move it forward.

--

Reviewer #3 (Remarks to the Author):

The authors have satisfactorily addressed my concerns.

We are glad that this reviewer is satisfied with the modifications introduced in the paper.

Note: In addition, we have introduced the editorial changes requested, as specified in file "CSantos et al Nat Comm Editorial issues response".